# A prefrontal cortex-lateral hypothalamus circuit controls stress-driven increased food intake

L. F. Supiot [1,2], K. L. Kooij[1], W. Du[1], A. A. C. Benschop[1], A. S. J. Nicolson [1], R. Haak [1], I. G. Wolterink-Donselaar[1], M. C. M. Luijendijk [1], D. Riga[1,2], R. A. H. Adan [1], R. B. Poorthuis[1,3] ✉ & F. J. Meye [1,3] ✉

Stress can drive overconsumption of high-fat foods. The medial prefrontal cortex (mPFC) is implicated in such stress-eating, but the underlying circuit mechanisms remain unclear. Here, we show that mPFC projections to the lateral hypothalamus (LHA) are required for stress-induced fat intake in male mice. We find that mPFC-LHA stimulation in sated states increases fat intake. Social stress acutely engages mPFC-LHA neurons, and inhibiting this pathway selectively prevents stress-driven excess fat intake. Circuit mapping shows that mPFC neurons innervate GABAergic and glutamatergic LHA ($LHA_{VGLUT2}$) neurons, but that social stress preferentially engages mPFC-$LHA_{VGLUT2}$ neurons and causes plasticity at mPFC-$LHA_{VGLUT2}$ synapses. Specifically, stress weakens mPFC synapses onto $LHA_{VGLUT2}$ neurons that curtail food intake, while strengthening mPFC synapses onto midbrain-projecting $LHA_{VGLUT2}$ neurons linked to stress-eating. We show that $LHA_{VGLUT2}$ neurons are required downstream mPFC targets for transforming stress into heightened fat intake. Overall, we identify the mPFC-LHA as a multi-branched network, indispensable for stress-eating.

Stressful experiences drive excessive intake of rewarding foods rich in fat and/or sugar[1–3]. In vulnerable individuals, stress contributes to the development of obesity, or to recurrent binge eating in Bulimia Nervosa (BN) and Binge Eating Disorder (BED)[4,5]. The prefrontal cortex (PFC), a stress-sensitive region important in decision making and strategy selection[6,7], is likely implicated in stress-driven food intake. In obese individuals and patients with BED/BN the PFC is hyperresponsive to food cues[8,9], and rodent studies have also demonstrated that stimulation of subsets of medial (m)PFC neurons can cause voracious food intake[3,10,11]. Despite its links to regulating eating behavior, it remains unknown whether the mPFC also transforms stressful experiences into excessive intake of palatable food, and if so, through which network adaptations.

A likely downstream target of the mPFC is the lateral hypothalamic area (LHA). mPFC glutamatergic pyramidal neurons send direct projections to the LHA[12]. The LHA itself has a prominent role in food intake[13–16], and we recently showed that the LHA interacts with the ventral tegmental area (VTA) in the midbrain to drive stress-driven food intake[3]. Notably, however, the LHA is highly heterogeneous in (functional) neuronal subtypes[3,14,15,17,18]. In particular, glutamatergic LHA neurons ($LHA_{VGLUT2}$ cells) are commonly linked to curtailing food intake[15,17], although there are also glutamatergic LHA neuronal subtypes, including those that project to the VTA, that can, under certain conditions, promote food intake[3]. Furthermore, certain neuropeptidergic subsets of glutamatergic LHA neurons (e.g., orexin, MCH), as well as GABAergic LHA neurons ($LHA_{VGAT}$ cells) can also promote food intake[15,17,19]. It is unclear which of these distinct LHA subtypes are innervated by mPFC and how mPFC control, potentially over distinct LHA cell types, orchestrates (stress-driven) feeding responses.

[1]Department of Translational Neuroscience, Brain Center, UMC Utrecht, Utrecht University, Utrecht, The Netherlands. [2]Department of Human Genetics, Center for Neurogenomics and Cognitive Research, Amsterdam University Medical Center, Amsterdam, The Netherlands. [3]These authors contributed equally: R. B. Poorthuis, F. J. Meye. ✉e-mail: rogier.poorthuis@gmail.com; f.j.meye-2@umcutrecht.nl

We hypothesized that stress would alter mPFC control over LHA networks, by activating LHA neuronal sets linked to promoting food intake, and/or by inhibiting neuronal sets linked to curtailing feeding, to causally drive excessive palatable food intake. Here we show, in mice, that optogenetic stimulation of the mPFC-LHA pathway is capable of driving intake of fat over chow. We use in vivo electrophysiology and fiber photometry to demonstrate that mPFC cells projecting to the LHA are directly sensitive to social stress and are engaged during fat intake. We demonstrate that mPFC-LHA pathway activity, while not important for regular feeding, becomes indispensable for the hyperphagia occurring after social stress. We investigated the downstream targets of mPFC, by combining patch clamp and optogenetics, showing that mPFC cells make direct synapses onto multiple LHA cellular subtypes. In the aftermath of stress, when the drive for food intake is increased, we observe weakening of mPFC glutamatergic synapses onto LHA$_{VGLUT2}$ cell types linked to reducing food intake but strengthening of mPFC inputs onto LHA$_{VGLUT2}$-VTA cells, which we previously implicated in mediating stress-driven binge responses[3]. Using in vivo optogenetics, we show that these divergent stress-driven circuit modifications increase the efficacy with which the mPFC-LHA network can persistently drive fat intake. Finally, we use circuit and neuronal ensemble-based approaches to identify that LHA$_{VGLUT2}$ neurons downstream of the mPFC, are indispensable integrators of stress information and subsequent hyperphagia. Overall, we identify the mPFC as a direct regulator of multiple LHA feeding circuits, with a critical role of this network in stress-driven excess intake of palatable fat.

## Results

### mPFC-LHA stimulation, in a frequency-dependent manner, increases palatable food intake

We first assessed whether the mPFC-LHA pathway is capable of driving intake of palatable food, as thus far this had not been reported. Instead, prior studies evaluating the mPFC-LHA pathway showed that high-frequency optogenetic stimulation (100 Hz bursts) does not affect regular food intake[20], while continuous chemogenetic stimulation (or 20 Hz optogenetic stimulation) decreases chow intake[21]. Notably, both mPFC and LHA are individually known to have frequency-dependent functions, with lower frequency stimulation able to drive distinct behaviors[19,22]. Therefore, we tested whether low-frequency stimulation of the mPFC-LHA pathway can increase palatable food intake. We stereotactically injected C57B6 mice in the mPFC with an AAV to express the optogenetic construct CoChR (rAAV5-Syn-CoChR-GFP) or YFP control vector (rAAV5-Syn-eYFP), and targeted optic fibers to the LHA to optogenetically stimulate the mPFC-LHA pathway (Fig. 1A, B; Supplementary Fig. 1A). After recovery, mice were placed in a 2-choice food-context containing regular chow and fat as a palatable food source. Baseline food intake was established over 3 consecutive days, followed by a session during which the mPFC-LHA pathway was stimulated at a specific frequency (Fig. 1C). We observed that during sessions of 5 Hz stimulation, but not of 1 or 10 Hz, there was an overall increase of fat intake (Fig. 1D; Supplementary Fig. 1B). Between the two choices, this increased intake was specific to fat, as it did not occur for chow (Supplementary Fig. 1C).

We then used video-based analysis to examine the feeding behavior within mPFC-LHA stimulation blocks (ON), contrasting them with behavior during alternating non-stimulated blocks (OFF). We observed that with 5 Hz stimulation, mice spent more time specifically in the fat zone (Supplementary Fig. 1D). Furthermore, we observed that the effect of pathway stimulation on fat consumption differed over the course of the task. Interactions with fat normally waned over the hour (Fig. 1E). Whereas 5 Hz stimulation of the pathway did not affect fat intake at the start, when the drive for fat was still high, it instead drove continued fat consumption later on when such interactions had typically waned, as seen in the control groups and in the 1 Hz stimulation

conditions (Fig. 1E). Consequently, 5 Hz stimulation resulted in overall more feeding bouts for fat, but not chow (Fig. 1F; Supplementary Fig. 1E). These effects on food intake did not stem from effects of pathway stimulation on general locomotor activity, which increased linearly in a different pattern from the effect on food intake (Supplementary Fig. 1F), nor did they stem from any direct alterations of emotional valence states, as assessed with a real-time place preference/avoidance assay (Supplementary Fig. 1G). Overall, these data indicate that the mPFC-LHA pathway is able, at least in the lower frequency domain, to drive the continued intake of palatable fat.

### mPFC cells projecting to LHA modulate their firing in response to social stress

Stressful experiences can drive excessive food intake (i.e., hyperphagia)[3,23], ultimately causing weight gain[23]. We reasoned that if the mPFC-LHA network were involved in stress-driven effects on food intake, it should be able to detect stressful information. To test this hypothesis, we used a social stress experience, where a mouse is temporarily exposed to the aggression and proximity of a dominant conspecific. We and others have previously shown that hours to days after such social stress, there is an increased intake of high palatable foods such as fat[3,23].

We assessed whether social stress would immediately engage the mPFC-LHA pathway. To this end, we performed in vivo electrophysiological recordings from mPFC cells projecting to the LHA. We injected, in C57B6 mice, a canine adeno virus in the LHA to retrogradely deliver Cre (Cav2-Cre) to the mPFC, and an AAV in the mPFC to Cre-dependently drive an opsin (AAV-DIO-hChR2(H134R)-mCherry). We then chronically implanted electrodes together with an optic fiber (i.e., an optrode) in the mPFC of these mice (Fig. 2A, B). This allowed for optogenetic-assisted identification (i.e., opto-tagging) of mPFC cells that projected to the LHA (Fig. 2C; Supplementary Fig. 2A, B). We then exposed these mice to the sensory proximity of a CD1 mouse and the actual fight experience with such an aggressor, while performing single-cell recordings from opto-tagged mPFC-LHA cells (Fig. 2D). We analysed firing rates of opto-tagged neurons before, during, and after the period in which fighting took place. Divergent responses were observed in mPFC neurons projecting to LHA during stress, with approximately half of the cells decreasing their firing rates during fight periods, whereas others were not affected or showed increased firing rates (Fig. 2E, F). These effects were not solely linked to the physical altercation, as they outlasted the fighting period (Fig. 2E; Supplementary Fig. 2C, D). In further support of this, we also evaluated the responses of these same neurons when the CD1 mouse arrived in the cage, while still behind a semi-permeable barrier (i.e., sensory stress). We observed similar response types, where in particular the neurons that increased their firing during sensory stres, were modulated in the same direction during physical exposure (Supplementary Fig. 2E–H). Aside from evaluating opto-tagged mPFC neurons with confirmed projections to the LHA, we also evaluated putative pyramidal mPFC neurons that were not opto-tagged. Such neurons also responded with excitations or inhibitions to social stress (Supplementary Fig. 2I, J), although they were less often modulated by stress than the mPFC-LHA population (Supplementary Fig. 2I). Overall, these data indicate the existence of different populations of mPFC-LHA neurons, with certain mPFC-LHA neurons acutely decreasing their activity, and others directly increasing it upon social stress exposure. These findings confirm the hypothesis that mPFC cells that project to LHA are directly able to detect a social stressor with known hyperphagic properties[3,23].

### mPFC-LHA pathway activity is indispensable for stress-driven excess fat intake

We showed that activity of the mPFC-LHA pathway is modulated by a social stressor. Next, we asked whether activity in this pathway is

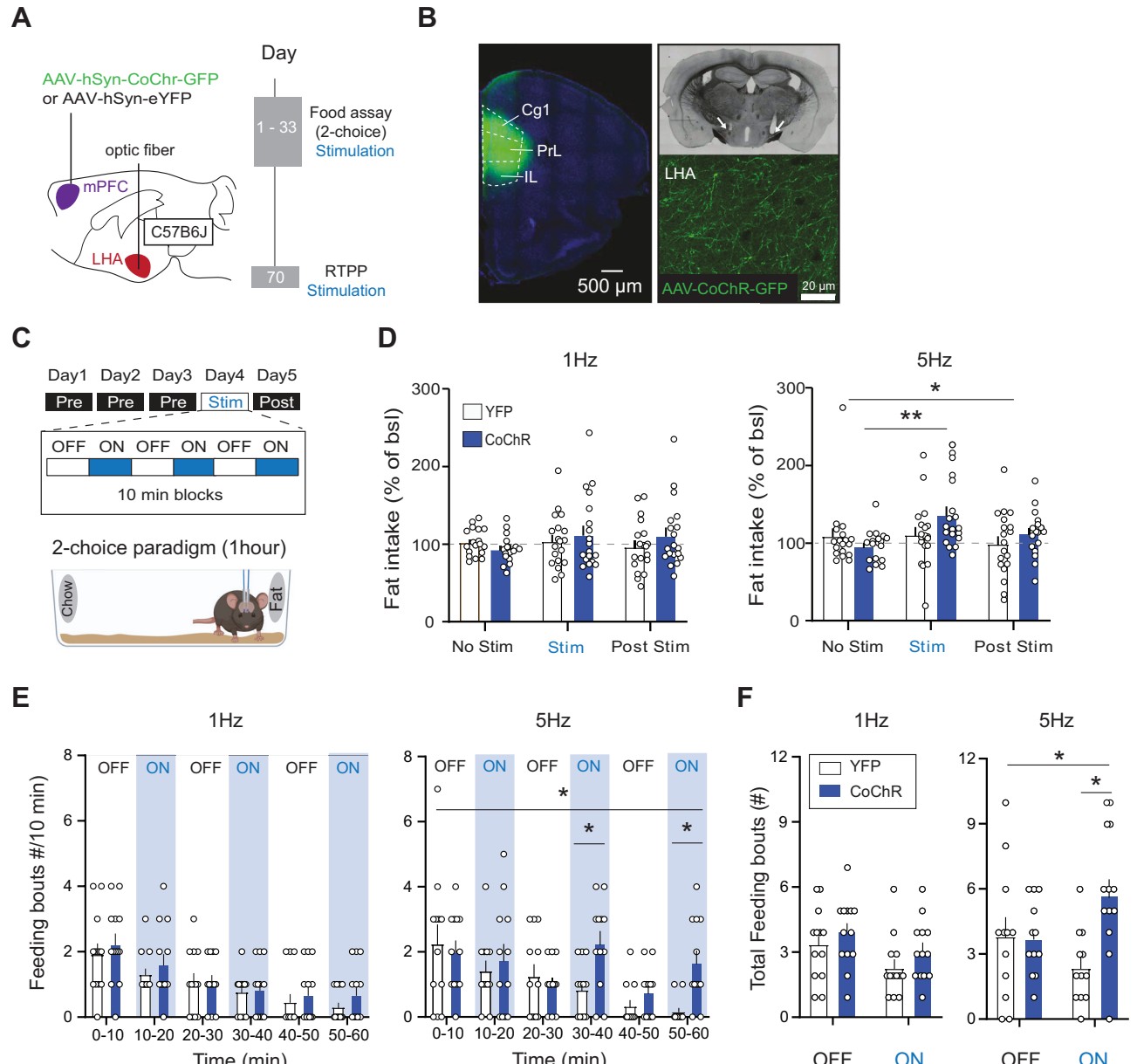

**Fig. 1 | Frequency-dependent increase of palatable food intake by mPFC-LHA pathway stimulation. A** Experimental timeline and viral and optogenetic strategy to manipulate mPFC-LHA pathway activity. **B** Left: Histological verification of CoChR-GFP expression in mPFC subterritories, cingulate cortex (Cg1), Prelimbic cortex (PrL), and infralimbic cortex (IL) (top), and optic fiber location in LHA (bottom, white arrows indicate end of optic fibers). Right: mPFC axons in LHA. **C** Schematic of behavioral paradigm. Food intake on the test day was measured against three baseline days. On stimulation day mPFC-LHA pathway was stimulated in blocks of 10 min alternated by non-stimulated blocks. **D** Bar graphs showing fat consumption normalized to intake of the first three baseline days for different stimulation frequencies. Plots show fat intake on baseline day 3, stimulation day and post-stimulation day. For 1 Hz: YFP $n = 19$, CoChR $n = 19$, Two-way RM ANOVA, Day-Virus interaction $F_{(2,72)} = 1.34$, $p = 0.18$. For 5 Hz: YFP $n = 19$, CoChR $n = 19$, Two-way RM ANOVA, Day-Virus interaction, $F_{(2,72)} = 3.37$, $p = 0.04$. Opsin-expressing mice on stimulation day compared to baseline (YFP $n = 19$, CoChR $n = 19$, Sidák's multiple comparison test, $p = 0.007$). **E** Bar graphs of video-based analysis of feeding bouts over the hour without or with mPFC-LHA optogenetic stimulation. Effect of 1 Hz (left): YFP $n = 13$, CoChR $n = 13$, Two-way RM ANOVA, Interaction Time-Virus, $F_{(5,120)} = 0.17$, $p = 0.95$. Effect of 5 Hz (right): YFP $n = 12$, CoChR $n = 12$, Two-way RM ANOVA, Interaction Time-Virus, $F_{(5,110)} = 2.83$, $p = 0.037$. Sidák's multiple comparison test 30-40 m, YFP vs CoChR, $p = 0.038$, and 50−60 m, $p = 0.013$. **F** Bar graphs for the effect of mPFC-LHA stimulation on feeding bouts quantified according to laser ON and OFF blocks. For 1 Hz: YFP $n = 13$, CoChR $n = 13$, Two-way RM ANOVA, Interaction Time-Virus, $F_{(1,24)} = 0.06$. For 5 Hz: YFP $n = 12$, CoChR $n = 12$, Two-way RM ANOVA, Interaction Time-Virus, $F_{(1,22)} = 7.64$, $p = 0.011$, Sidák's multiple comparison test ON blocks YFP vs CoChR, p = 0.003). In bar graph (**D**–**F**) the superimposed points represent individual animals and error bars represent their SEMs. *$p < 0.05$; **$p < 0.01$.

necessary for stress-driven increases in food intake. To this end, we bilaterally expressed an inhibitory chemogenetic receptor in the mPFC-LHA pathway in C57B6 mice. Specifically, we stereotactically injected Cav2-Cre retrograde virus in the LHA, and an AAV to Cre-dependently drive hM4Di (or a control fluorophore) in the mPFC

(Fig. 3A, B). We confirmed that this indeed resulted in chemogenetic control over LHA-projecting mPFC cells. In brain slices, we patch-clamped such mPFC neurons and found that bath administration of DREADD agonist CNO resulted in strong hyperpolarization of the neurons (Supplementary Fig. 3A).

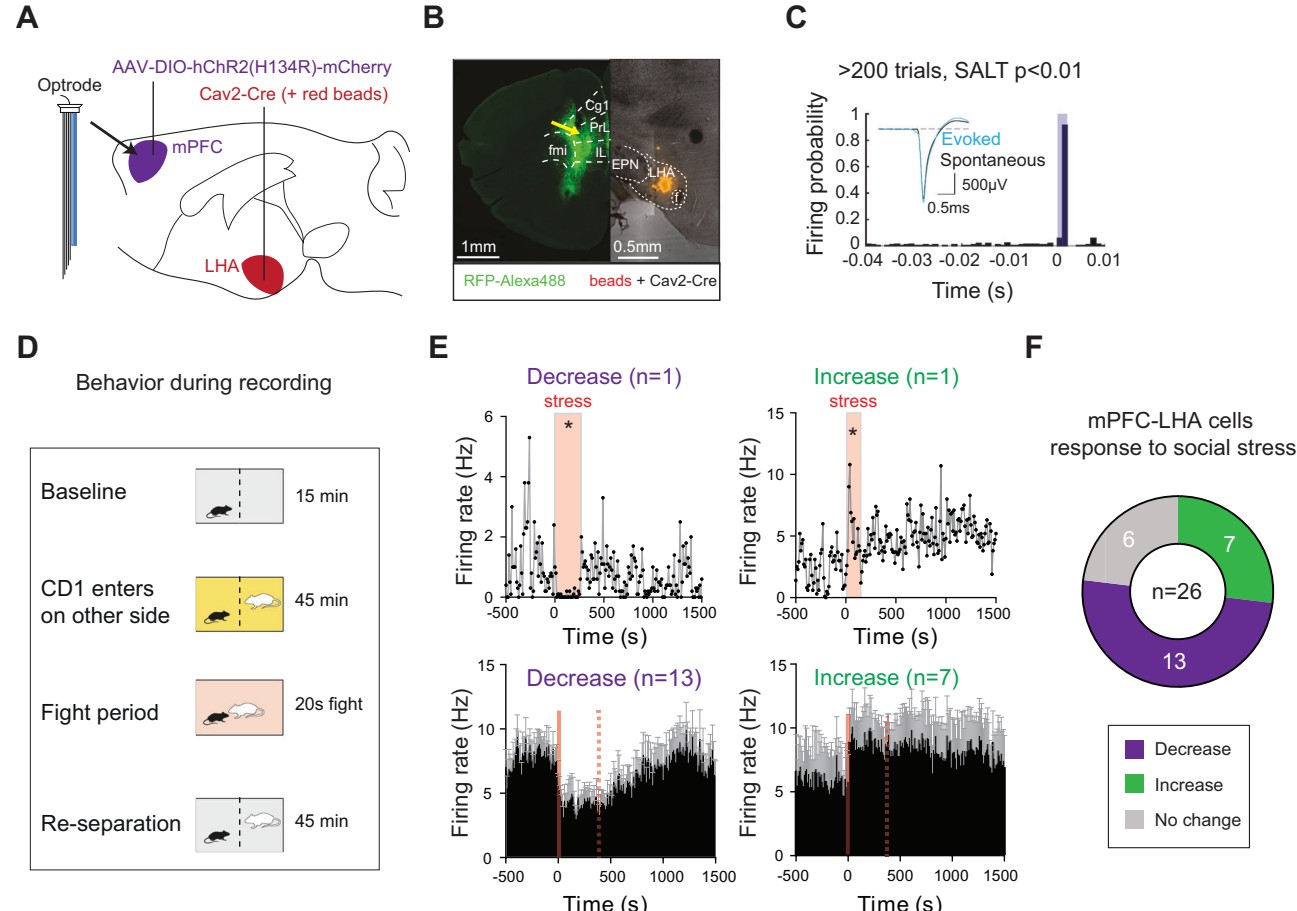

**Fig. 2 | mPFC-LHA neuronal activity is diversely modulated by social stress.**
**A** Viral strategy and implantation of optrodes (i.e., 32 electrode wires and an optic fiber) for extracellular recordings of opto-identified LHA-projecting mPFC neurons. **B** Left: Histological verification of placement of electrode tip in between opsin-expressing mPFC neurons projecting to LHA. Tip of optrodes is indicated with a yellow arrow. Right: Histology of LHA targeting with the Cav2-Cre retrograde virus. Anatomical reference point: fmi=forceps minor of the corpus callosum, EPN=entopeduncular nucleus, f = fornix. **C** Optogenetic-assisted identification of mPFC pyramidal neurons projecting to the LHA. Neurons were classified as being opto-tagged if they showed a significant increase in firing rate after light stimulation compared to baseline using the SALT-test (see methods for test and classification

criteria, and see Suppl. Datafile for individual neuron statistics). **D** Behavioral paradigm to record mPFC-LHA cellular activity during social stress. **E** Bidirectional modulation of distinct mPFC-LHA neurons by social stress. Top: histograms of single cell examples (10 s bins comparing 180 s right before and after fight onset, Wilcoxon signed-rank test, $p < 0.05$). Bottom: Histograms showing group averages ($n = 26$ opto-identified mPFC-LHA cells from 12 mice; mPFC-LHA cells that decrease firing $n = 13$ cells; mPFC-LHA cells that increase firing $n = 7$ cells). The dotted line indicates the average length of the stress exposure. Error bars in histograms indicate SEMs across neurons. **F** Pie chart showing mPFC-LHA neurons, which were significantly modulated by, or not responsive to, the social stress experience ($n = 26$ opto-tagged mPFC-LHA cells from 12 mice). *$p < 0.05$.

We then tested whether chemogenetic inhibition of the pathway would alter regular or stress-driven feeding behavior. We again placed animals in the 2-choice model (as in Fig. 1C) during a baseline session. Then we exposed the mice to either two days of social stress (which we previously showed to increase fat intake[3]) or to control exposure. The next day, we systemically administered CNO to the animals, and 45 minutes later reintroduced them to the 2-choice model (Fig. 3C). Mice with control mCherry vectors in the mPFC-LHA pathway exhibited clear stress-driven increased intake compared to their baseline. In mice with hM4Di in their mPFC-LHA pathway, we observed that inhibiting the pathway had no effect on food intake for non-stressed mice. Instead, in stressed mice it completely counteracted the amount of overconsumption that occurred after stress (Fig. 3C, D, Supplementary Fig. 3B). The effects of mPFC-LHA pathway inhibition were not due to actions on locomotor behavior or altered anxiety (Supplementary Fig. 3C, D). No differences were observed for chow intake (Fig. 3D). Overall, this suggests that mPFC-LHA pathway activity is not required for regular food intake, but becomes indispensable for regulating the excess food intake after stress.

## mPFC neurons projecting to LHA respond to food intake

We next investigated whether the mPFC-LHA pathway would be responsive to food intake and stress-modulations thereof. We investigated the mPFC-LHA pathway in this case by measuring population-level calcium responses using fiber photometry. We stereotactically injected Cav2-Cre in the LHA, injected an AAV for Cre-dependent expression of calcium indicator GCaMP8s in the mPFC, and placed a fiber for photometric measurements above it (Fig. 3E; Supplementary Fig. 3G). Our in vivo electrophysiology data had shown that social stress is a modulating experience for this pathway, resulting at the single cell level at either excitation or inhibition in a considerable subset of cells (Fig. 2E, F). When we exposed mice to the social aggression of a CD1 mouse, we observed a biphasic calcium response of the mPFC-LHA at the population level to this stimulus, with a peak and a subsequent dip in calcium responses (Fig. 3F, G; Supplementary Fig. 3H). This confirms that with fiber photometric measurements, we can detect relevant dynamics in this pathway.

We then evaluated the response of the mPFC-LHA pathway to food intake. For that, we placed animals in the 2-choice task with

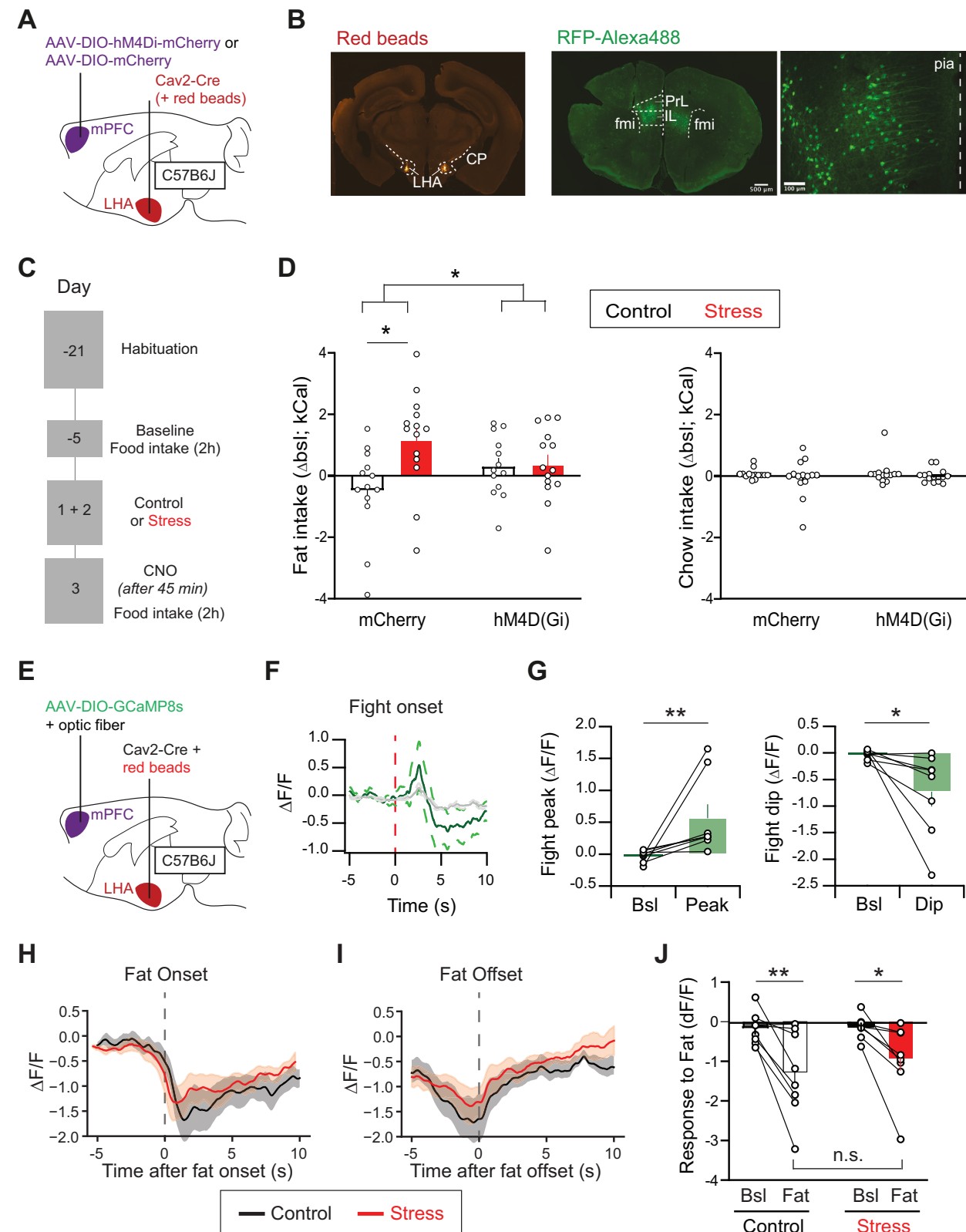

access to fat and chow. We observed that when mice ate, and particularly when they ate fat, there were pronounced dips in calcium signals recorded from mPFC neurons projecting to LHA (Supplementary Fig. 3I). These were time-locked to the onset of food contact and waned after disengagement from the food source (Supplementary Fig. 3I). We then evaluated whether such responses, measured from mPFC somata, would be altered by prior stress. We exposed the mice

again to a two-day social stress paradigm (or to control). We then recorded the day after, where normally there would be an increased food intake (Fig. 3D). However, we observed the same type of feeding-related dip responses in the pathway in control and in stressed mice (Fig. 3H–J). Hence, although the mPFC-LHA pathway is modulated during both stress and during feeding responses, and stress alters feeding responses, we did not observe signs of stress-driven plasticity

**Fig. 3 | mPFC-LHA neuronal activity is required for stress-driven excessive intake of fat and the pathway is modulated by feeding. A** Viral strategy to make mPFC-LHA neurons susceptible to chemogenetic inhibition. **B** Histology of injection sites of CAV2-Cre in the LHA using red beads, and of hM4Di-mCherry expression in mPFC subterritories PrL and IL (antibody for green fluorescence). Anatomical reference point: cp=cerebral peduncle. **C** Timeline of the stress-driven food intake experiment with chemogenetic mPFC-LHA manipulation. **D** Effect of chemogenetic inhibition of mPFC-LHA on food intake. Delta food intake during test day (Day +3) compared to the baseline session (Day −5). Left: Effect on (stress-driven) fat intake (mCherry-Control $n = 13$, hM4Di-control $n = 13$, mCherry-Stress $n = 14$, hM4Di-Stress $n = 14$, Two-way ANOVA, Treatment-Virus interaction, $F(1,50) = 4.69$, $p = 0.035$, Sidáks multiple comparison test mCherry-control vs mCherry-Stress, $p = 0.015$, hM4Di-control vs hM4Di-Stress, $p > 0.99$, mCherry-control vs hM4Di-control, $p = 0.61$). Right: Effect on chow intake (mCherry-Control $n = 13$, hM4Di-control $n = 13$, mCherry-Stress $n = 14$, hM4Di-Stress $n = 14$, Two-way ANOVA, Treatment-Virus interaction, $F(1,50) = 0.07$, $p = 0.79$). **E** Schematic and timeline for fiber photometric measurements from mPFC neurons projecting to LHA. **F** Line graph showing average calcium response time course of the mPFC-LHA population to social stress. Dotted lines show SEMs, based on different mice ($n = 8$). **G** Left: Bar graph quantifying the peaks after social stress. $N = 8$ mice, Wilcoxon matched pairs, $W = 36$, $p = 0.008$. Right: Bar graph quantifying the dips after social stress. $N = 8$ mice, Wilcoxon matched pairs, $W = 34$, $p = 0.016$. **H** Line graph showing time courses of dips in mPFC-LHA pathway during onset of food intake in either control context ($n = 8$ mice) or after 2 days of social stress ($n = 8$ mice). Shaded areas show SEMs, based on different animals. **I** As **H** but for the offset of fat interaction. **J** Bar graph quantifications of dips with respect to baseline, both in control ($n = 8$) or after social stress ($n = 8$). Two-Way ANOVA, Stress-Time interaction, $F(1,14) = 0.53$, $p = 0.48$. Main effect Time ($F(1,14) = 21.01$, $p = 0.0004$. Fisher's LSD multiple comparison control baseline vs fat contact $t(14) = 3.754$, $p = 0.0021$, stress baseline vs fat contact $t(14) = 2.729$, $p = 0.0163$. In bar graph (**D**, **G**, **J**) superimposed points represent individual animals and error bars represent SEMs. *$p < 0.05$; **$p < 0.01$.

in LHA-projecting mPFC soma reactivity during stress-driven hyperphagia. We next investigated whether stress-driven plasticity would instead occur downstream at mPFC synapses onto LHA neurons.

## mPFC neurons make monosynaptic connections to multiple types of LHA neurons

Our findings demonstrate that the mPFC neurons projecting to the LHA respond to social stress and are capable of altering food intake, respond to food intake, and subsequently are necessary for stress-driven overconsumption of fat. However, no changes were observed in the somatic population activity of mPFC-LHA cells after stress while interacting with palatable food. This led us to hypothesize that plasticity of downstream communication with different populations in the LHA might underlie alterations in fat intake. However, it is unclear which LHA neuronal subtypes are innervated by the mPFC. The LHA contains both glutamatergic populations, expressing the vesicular glutamate transporter 2 ($LHA_{VGLUT2}$) and GABAergic populations, identified by expression of the vesicular GABA transporter ($LHA_{VGAT}$). Recent transcriptomic evidence suggests that essentially all LHA neurons can be ascribed to one of these major classes, with further key subpopulations occurring within them[15,18]. For instance, evidence suggests that orexin ($LHA_{OREX}$) and melanin-concentrating hormone ($LHA_{MCH}$) releasing neurons represent distinct subsets of LHA glutamatergic (VGLUT2+) neurons, but also that most $LHA_{VGLUT2}$ neurons do not belong to those specific subsets[18]. We reasoned that, despite further heterogeneity within $LHA_{VGAT}$, $LHA_{VGLUT2}$, $LHA_{MCH,}$ and $LHA_{OREX}$ neuronal sets[15,18], these four populations offer a starting point to understand mPFC functional innervation patterns of the LHA.

To map mPFC connectivity to these four LHA subpopulations, we performed an optogenetic-assisted synaptic connectivity study[24,25]. We targeted the mPFC with the opsin CoChR (AAV5-Syn-CoChR-GFP) and fluorescently labeled $LHA_{VGLUT2}$, $LHA_{VGAT}$, $LHA_{MCH}$ or $LHA_{OREX}$ populations with viral and transgenic strategies. For $LHA_{VGLUT2}$ and $LHA_{VGAT}$ populations we used well-established transgenic mice VGLUT2-Cre or VGAT-Cre mice, respectively (Fig. 4A, B). We injected these mice in the LHA with an AAV to Cre-dependently express a red fluorophore (rAAV-EF1a-DIO-mCherry) to label either $LHA_{VGLUT2}$ or $LHA_{VGAT}$ neurons, respectively (Fig. 4A, B). Instead, to target $LHA_{MCH}$ or $LHA_{OREX}$ neurons we injected C57B6J mice with an AAV to drive expression of a red fluorophore under the control of short promotors for either MCH (AAV-MCHpr-Gq-mCherry) or orexin (AAV-hORXpr-TdTomato)[26] (Fig. 4A, B). We established the specificity of this approach using post-hoc labeling of biocytin-filled patched cells expressing either the fluorophore under control of the MCH or the orexin promotor (Supplementary Fig. 4A–C). We also established that the $LHA_{VGLUT2}$ cells that we on average targeted were distinct from the $LHA_{OREX}$ and $LHA_{MCH}$ cells, as those peptidergic subpopulations of glutamatergic neurons differed strongly in somatic cell size from typical $LHA_{VGLUT2}$

neurons (Supplementary Fig. 4D). The four distinct LHA populations also differed from each other in other biophysical properties, further corroborating that different LHA populations were sampled (Supplementary Fig. 4E–I). We also established that both $LHA_{VGAT}$ and $LHA_{VGLUT2}$ neurons were able to follow protracted optogenetic stimulation of mPFC inputs at least for 1 and 5 Hz, whereas for 10 Hz fidelity of induced postsynaptic potentials remained higher onto LHA glutamatergic neurons (Supplementary Fig. 4J).

We then performed ex vivo patch clamp recordings to assess the occurrence of mPFC synaptic inputs onto these LHA subpopulations. We observed high degrees (≥80%) of connectivity between mPFC cells and all four LHA populations (Fig. 4C). We then repeated these experiments in the presence of a pharmacological cocktail of tetrodotoxin (TTX) and 4-aminopyridine (4-AP) in the external medium[24,27] and confirmed that the great extent of connectivity of mPFC inputs with all LHA subtypes was due to direct monosynaptic connections (Fig. 4D). Together, these data indicate that there are strong monosynaptic projections from mPFC cells to all four LHA subpopulations.

## Stress selectively affects mPFC synaptic contacts onto $LHA_{VGLUT2}$ neuronal populations via presynaptic processes

After social stress, there is an increased drive for the intake of palatable food sources rich in sugar and fat[3,23], which ultimately results in excess body weight if maintained over weeks[23]. In the fiber photometric experiment, we observed that at the level of mPFC cell bodies, the two-day stress model did not affect response magnitudes. However, it remains possible that downstream, at the synaptic level, changes occurred due to stress. We therefore next asked if, at the time of stress-driven hyperphagia, mPFC synaptic strength onto distinct LHA neuronal populations would be altered. To address this, we evaluated the effects of stress on mPFC synaptic contacts onto the four distinct LHA populations ($LHA_{VGLUT2}$, $LHA_{VGAT}$, $LHA_{MCH}$, $LHA_{OREX}$). We expressed an opsin in the mPFC (AAV-CoChR-GFP) and tagged distinct LHA populations as before (Fig. 4A, B; Supplementary Fig. 5A, B). We exposed mice to two days of control experience or social stress, and then the next day, when mice normally engage in higher intake of palatable food[3] (Fig. 3D), we made brain slices and assessed synaptic properties at mPFC synapses onto the four broad classes of LHA neuronal populations (Supplementary Fig. 5A–C).

Interestingly, we observed that specifically at the mPFC synapses onto $LHA_{VGLUT2}$ neurons, there was, on average, a stress-induced decrease in maximally optogenetically-evoked EPSC magnitudes (Fig. 5A; Supplementary Fig. 5C). This suggests a weakening of glutamatergic mPFC synapses onto many $LHA_{VGLUT2}$ neurons. To further address the underlying mechanisms for this, we assessed the paired pulse ratio (PPR), a measure inversely related to presynaptic release probability[24]. We observed that

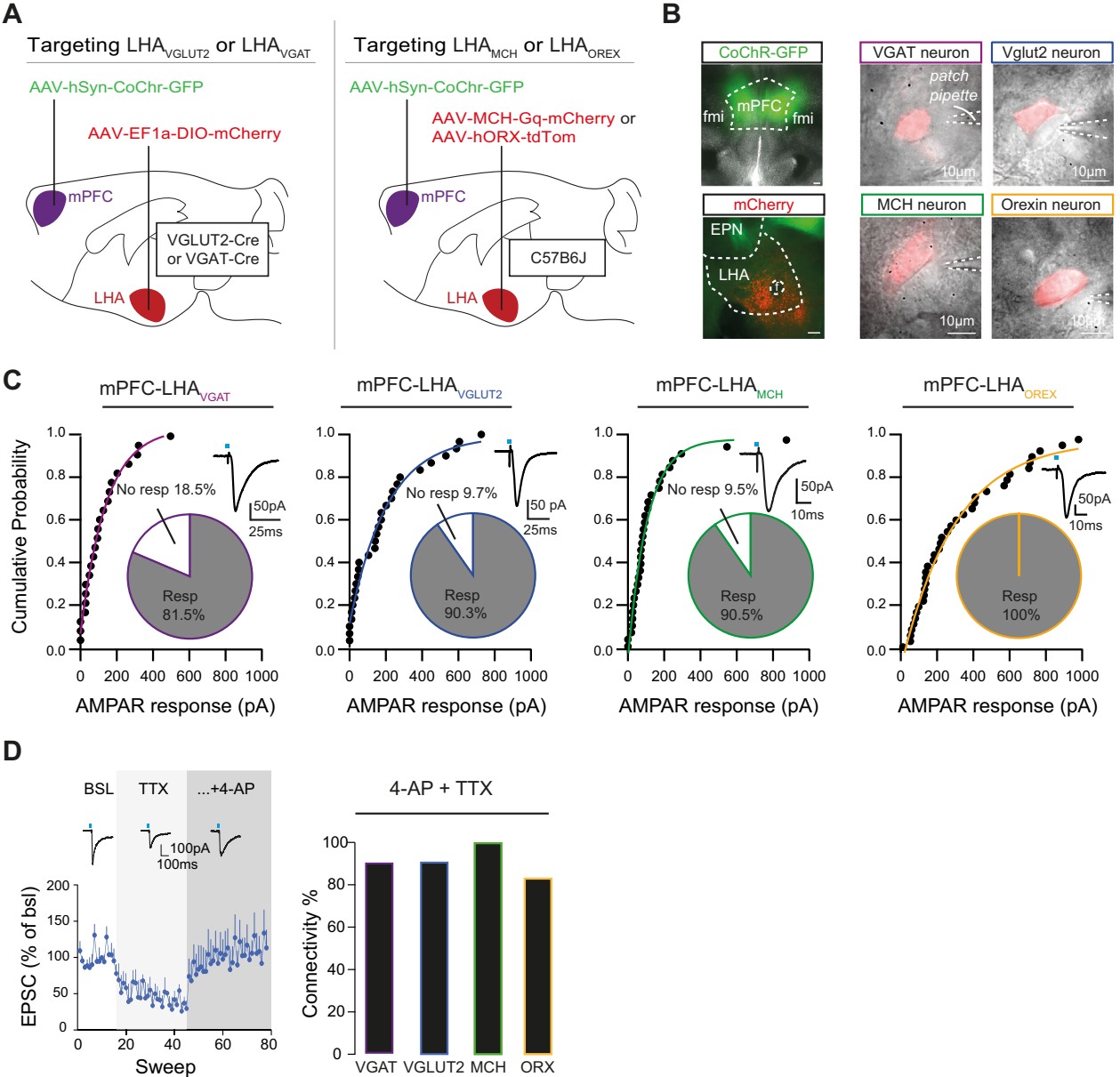

**Fig. 4 | mPFC sends direct excitatory projections to multiple LHA neuronal populations. A** Schematic of the viral strategies to identify LHA populations and to express an opsin in mPFC. **B** Left: Representative image of the viral expression in mPFC (in green) and in LHA cell types (in red) in coronal slices. Scale bar 200 μm. Right: Representative image of the different LHA neuronal subsets (red) during patch clamp experiments. Scale bar 10 μm. **C** Quantifications of extent of synaptic connectivity between mPFC and different LHA subpopulations. From left to right, cumulative probability plots of opto-evoked AMPAR amplitudes, after mPFC input stimulation, in LHA$_{VGAT}$ (n cells = 23 from 9 mice), LHA$_{Vglut2}$ (n cells = 30 from 10 mice), LHA$_{MCH}$ (n cells = 31 from 10 mice), and LHA$_{OREX}$ (n cells = 38 from 10 mice) populations. Alongside are pie chart representations of the proportion of LHA cells

connected to mPFC (n VGAT cells = 27, n Vglut2 cells = 31, n MCH cells = 32, and n Orexin cells = 38). For the voltage-clamp trace examples within the graph, blue rectangles represent 1 ms of opto-stimulation. **D** Left: Example timeline of EPSCs in an LHA$_{VGLUT2}$ cell during baseline, during application of TTX, and with further addition of 4-AP. SEMs represent variation in binned sweeps (bins of 60 s). Right: Bar graph showing the percentage of connectivity to mPFC for the 4 LHA populations in presence of 4AP + TTX (n VGAT cells = 10, 1 mouse, n Vglut2 cells = 11, 1 mouse, n MCH cells = 9, 2 mice and n Orexin cells = 12, 2 mice), showing a high monosynaptic connectivity that was similar across cellular populations (Pearson Chi-Square test, χ2 = 3.166, p = 0.367).

specifically at mPFC- LHA$_{VGLUT2}$ synapses there was an increased PPR, in accordance with a scenario of lower release probability at these synapses due to prior social stress (Fig. 5B; Supplementary Fig. 5D). The effect of stress was specific to the presynaptic compartment, as we did not observe a difference in postsynaptic glutamatergic receptor distributions at these synapses (Supplementary Fig. 5E). Nor did we observe changes due to stress in terms of the excitability of the LHA cells or in basic membrane properties (Supplementary Fig. 5F, G). These data suggest that

social stress alters mPFC glutamatergic synapses onto LHA$_{VGLUT2}$ neurons. We next asked whether another stressor would similarly be able to induce these effects. For another stress experience of a different modality we exposed mice to restraint stress (Supplementary Fig. 6A), and we observed that this also subsequently increased intake of fat (Supplementary Fig. 6B). Similarly to social defeat stress, restraint stress also reduced the magnitude of mPFC synaptic responses onto LHA$_{VLUT2}$ neurons (Supplementary Fig. 6C, D), and it also increased the PPR at these synapses,

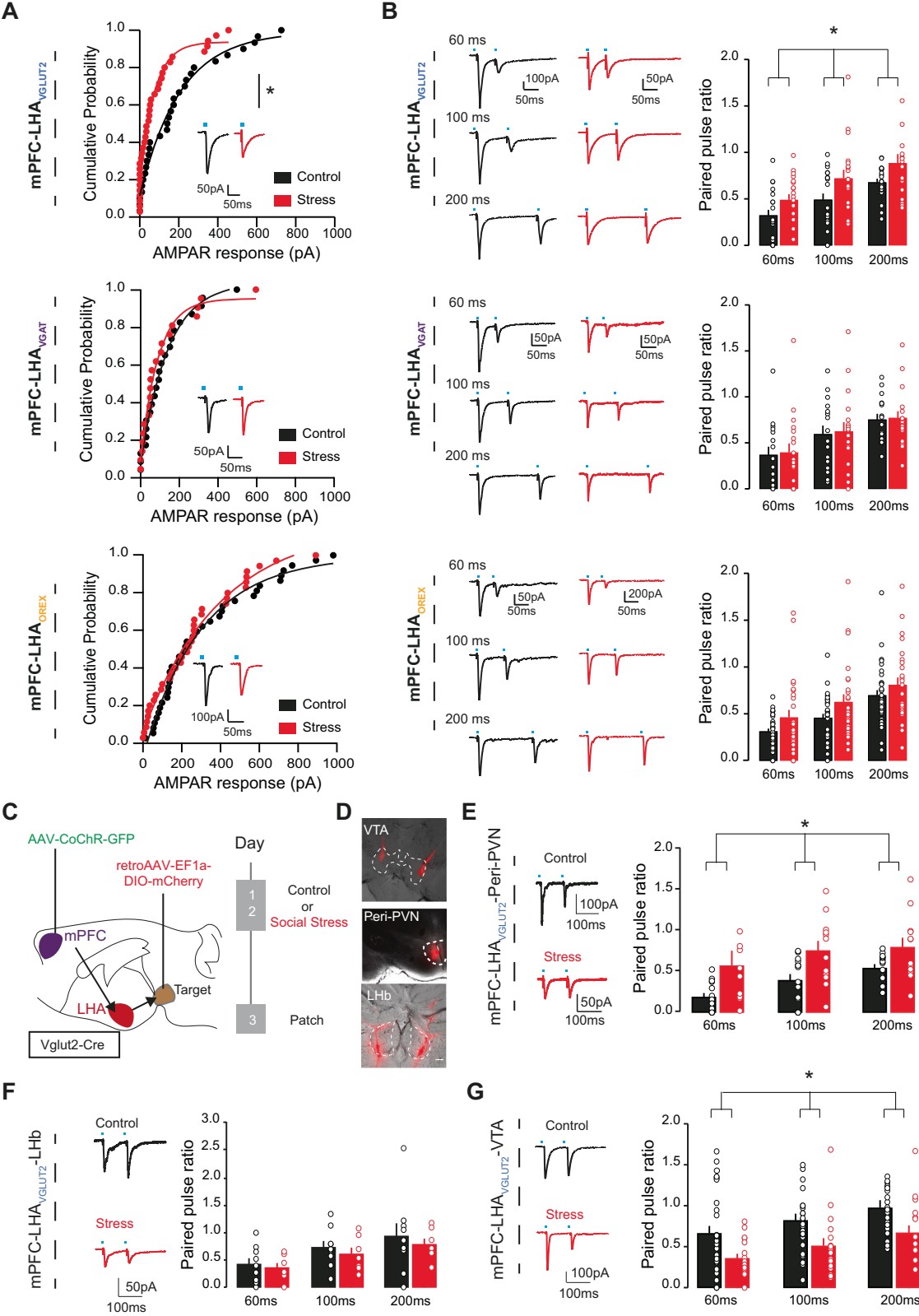

indicating a similar mechanism of reduced release probability between the different types of stressors (Supplementary Fig. 6E). Overall, these data suggest that after stressful experiences, which can drive binge intake of fat, there is on average a weakening of mPFC glutamatergic synapses onto the global population of LHA$_{VGLUT2}$ neurons, which on average reduces food intake upon activation[15,17].

## Social stress has divergent effects on distinct populations of mPFC-innervated LHA$_{VGLUT2}$ cells

LHA$_{VGLUT2}$ neurons still represent a very heterogeneous population[18] and in part this heterogeneity is reflected by their distinct projection targets[28]. For instance, stimulating LHA$_{VGLUT2}$ projections to the lateral habenula (LHb) suppresses food intake[29]. Similarly, stimulating LHA$_{VGLUT2}$ neuronal projections to midline hypothalamic regions

**Fig. 5 | Social stress drives divergent plasticity at mPFC synapses onto distinct LHA$_{VGLUT2}$ neuronal subsets. A** Cumulative probability plot of opto-evoked AMPARs in neurons from distinct LHA subpopulations for control (black) and stress (red) conditions, with representative example trace insets. Top: mPFC-LHA$_{VGLUT2}$ synapses (n control = 30, 10 mice, *n* stress = 35, 11 mice; Mann-Whitney test, U = 342, *p* = 0.0152). Middle: mPFC-LHA$_{VGAT}$ synapses (*n* control=23, 8 mice, n stress= 21, 9 mice; Mann-Whitney test, U = 232, *p* = 0.8341). Bottom: mPFC-LHA$_{OREX}$ population (n control = 37, 10 mice, *n* stress=35, 9 mice; Mann-Whitney test, U = 582, *p* = 0.4663). **B** Example traces (left) and bar graphs (right) for paired pulse ratios (PPRs) at mPFC synapses onto distinct LHA subpopulations, for control (black) and stress (red) conditions. PPRs at distinct inter-pulse intervals (IPIs) of 60, 100 and 200 ms are shown. Top: mPFC- LHA$_{VGLUT2}$ synapses (*n* control=19, 10 mice, *n* stress= 20, 11 mice; two-way RM ANOVA, Main effect of group, F(1,37) = 4.55, *p* = 0.04). Middle: mPFC-LHA$_{VGAT}$ synapses (n control= 16, 7 mice, n stress= 18, 9 mice; Two-way RM ANOVA, Main effect of group, F(1,32) = 0.160, p = 0.691).

Bottom: mPFC-LHA$_{OREX}$ synapses (*n* control = 32, 10 mice, n stress= 34, 9 mice; Two-way RM ANOVA, Main effect of group, F(1,64) = 1.752, *p* = 0.19). **C** Schematic and experimental timeline for patch clamp recordings of mPFC synaptic inputs on LHA$_{VGLUT2}$ cells with distinct downstream targets. **D** Representative images of retrograde targeting VTA, Peri-PVN, or LHb. Scale bar 200 μm. **E** Left: example traces for mPFC-LHA$_{VGLUT2}$-Peri-PVN synapses in control (black) and stress (red) conditions during 100 ms IPI. Right: Bar graph representation of PPRs (*n* control = 14, 5 mice, n stress = 12, 4 mice; two-way RM ANOVA, Main effect of group, F(1,24) = 5.975, p = 0.022). **F** As E, but for mPFC-LHA$_{VGLUT2}$-LHb synapses (*n* control= 9, 4 mice, n stress = 7, 4 mice; Two-way RM ANOVA, Main effect of group, F(1,14) = 0.099, *p* = 0.758). Scale bar 200 μm. **G** As E, but for mPFC-LHA$_{VGLUT2}$-VTA synapses (*n* control= 27, 9 mice, n stress=17, 6 mice; Two-way RM ANOVA, Main effect of group, F(1,42) = 7.13, *p* = 0.011). In bar graphs (**B**, **E**–**G**) superimposed points represent individual neurons, and error bars represent their SEMs. *$p$ < 0.05.

around the third ventricle (i.e., peri-paraventricular nucleus, or peri-PVN region[30]) is linked to aversive processes[31] and reduced food intake[32]. Instead LHA$_{VGLUT2}$ projections to the ventral tegmental area (VTA) do not alter food intake under naïve conditions[3,33,34], but the potentiation of their synapses onto dopamine neurons causally contributes to the enhanced fat intake occurring after social stress[3]. Given this LHA$_{VGLUT2}$ anatomical and functional divergence, we asked how social stress would affect mPFC synapses onto such distinct LHA$_{VGLUT2}$ populations. We first assessed if the LHb, VTA and (peri)-PVN could be downstream targets of mPFC-innervated LHA neurons. For this, we used a transsynaptic tracing strategy (Supplementary Fig. 6F) in which we injected AAV1-Cre in the mPFC, to cause a transsynaptic jump to downstream neurons[35]. We then injected an AAV-DIO-GFP in the LHA to perform anterograde tracing of the nerve terminals belonging to LHA neurons with mPFC-innervation (Supplementary Fig. 6F, G). We indeed observed anterograde terminals in the regions of interest, like the VTA, the LHb and the peri-PVN (Supplementary Fig. 6H).

We then investigated if and what type of stress-driven synaptic changes occurred at mPFC synapses onto LHA$_{VGLUT2}$ neurons projecting to LHb, VTA, and peri-PVN. To label LHA$_{VGLUT2}$ cell types based on their projection output, we injected a Cre-dependent retrograde tracer (retroAAV-DIO-mCherry) in Vglut2-Cre mice in these three different areas (Fig. 5C, D). We then again exposed mice to either two days of control or social stress conditions and we prepared brain slices the day after and patched from retrogradely traced LHA$_{VGLUT2}$ neurons, assessing their mPFC input (Fig. 5C). The extent of mPFC-innervation of the three types of LHA$_{VGLUT2}$ neurons with distinct projections was high (Supplementary Fig. 6I). We observed that mPFC synapses onto LHA$_{VGLUT2}$-Peri-PVN neurons exhibited higher PPR upon stress (Fig. 5E), in accordance with the general effect of stress on the LHA$_{VGLUT2}$ neurons (Fig. 5B). We confirmed, via in vivo optogenetic stimulation of the LHA$_{VGLUT2}$ projection to the peri-PVN, that this pathway indeed reduces food intake upon activity (Supplementary Fig. 6J-L as previously suggested[32]). In contrast to the effect of stress on mPFC-LHA$_{VGLUT2}$-Peri-PVN synapses, we did not observe PPR changes at mPFC synapses onto LHA$_{VGLUT2}$-LHb neurons after stress (Fig. 5F). Interestingly, we observed that mPFC synapses onto LHA$_{VGLUT2}$-VTA neurons presented a lower PPR after stress, suggesting enhanced release probability of mPFC synapses onto those specific LHA$_{VGLUT2}$ neurons (Fig. 5G). In accordance, we had previously shown that stress-driven strengthening of LHA glutamatergic synapses onto VTA dopamine neurons is linked to stress-driven fat intake[3].

Overall, these data suggest that social stress has divergent effects on the strength of mPFC synapses onto distinct LHA$_{VGLUT2}$ populations. Weakening excitatory mPFC synapses onto LHA$_{VGLUT2}$ populations with a role in reducing food intake[32], while strengthening excitatory mPFC synapses onto LHA$_{VGLUT2}$ populations with a role in promoting food intake[3].

## After social stress, mPFC-LHA activity is more primed to drive feeding

We reasoned that the consequences of such stress-driven network modifications could prime the mPFC-LHA pathway to become even more capable of driving palatable food intake when stimulated.

To test this, we again optogenetically probed the mPFC-LHA pathway by stereotactically injecting C57B6J mice with an optogenetic construct (rAAV5-Syn-CoChR-GFP), implanting optic fibers bilaterally above the LHA (Fig. 6A, B). After recovery, we either exposed the mice to two days of social stress or housed them with a novel cage mate. Subsequently, we placed the animals in the 2-choice task, where they could consume fat or chow for 2 h. Mice were again exposed to recurring blocks of optogenetic stimulation of 5 Hz, or instead to mock stimulation (Fig. 6A). We observed that even though mice that had previously experienced stress already consumed more fat than control mice, optogenetic stimulation could drive their intake of palatable food even further, making these mice the largest consumers of fat (Fig. 6C), in the absence of effects on chow (Supplementary Fig. 7A).

The overall fat intake reflected the outcome at the end of the two-hour binge period. We next addressed, with more fine-grained temporal resolution, whether optogenetic stimulation was more efficient in driving fat intake in stressed mice as compared to control mice. We first aimed to better understand the development of fat intake under stress (without pathway stimulation). We observed that regular (not stimulated) stressed animals performed over 75% of their fat feeding bouts in the first 20 minutes of a session, and then strongly diminished interaction with the fat from thereon (Fig. 6D, E). This binge-like feeding pattern of fat for stressed mice was different from that of control mice, which had fewer immediate feeding bouts but continued having feeding bouts for a longer period (Fig. 6D, E). We then analyzed how, in these two distinct phases that we refer to as "Binge Phase" (i.e., 0–20 min) and "Post-Binge phase" (20–120 min), mPFC-LHA pathway stimulation affected fat interaction. Reminiscent of our previous finding of pathway stimulation early in the 2-choice session in naïve mice (Fig. 1E), we found that 5 Hz mPFC-LHA stimulation was not effective in the early "Binge Phase" of the session (Fig. 6F). Instead in the subsequent "Post Binge" phase 5 Hz stimulation particularly drove stressed mice more potently towards the fat, when they would normally be in more sated conditions (Fig. 6F). To investigate whether indeed the 5 Hz stimulation could prevent a decline of interest for the fat, we plotted the relative drive for fat eating over the course of the task by comparing the relative amount of feeding bouts taking place during the binge and post-binge period. Indeed, particularly in stressed mice, 5 Hz stimulation did not affect the binge intake, but it did counteract the normally occurring decline of interest in the fat later on (Fig. 6G). Overall, these data reveal that after stress mPFC-LHA network activity is primed more towards an ability to drive continued fat intake.

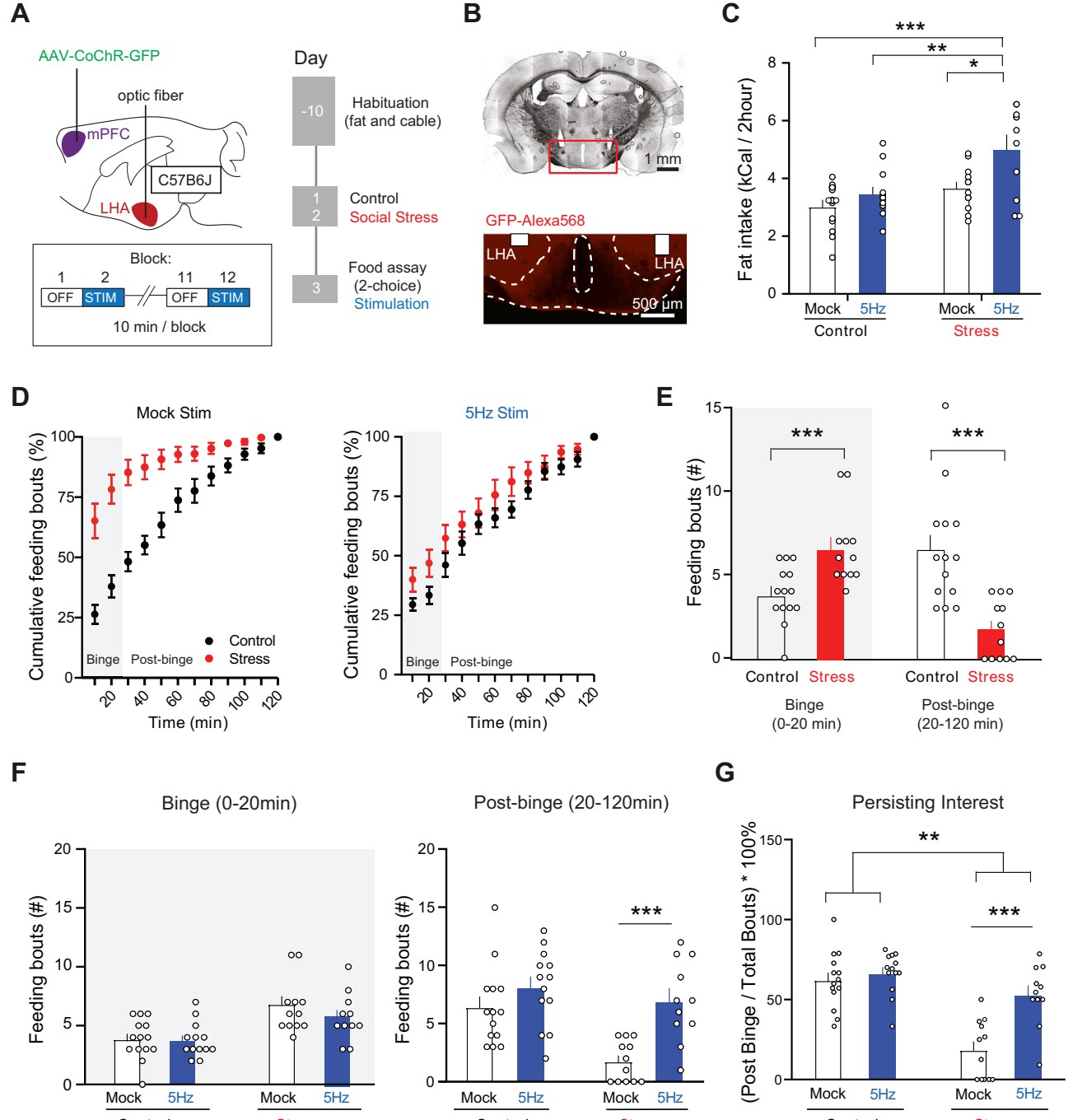

**Fig. 6 | In vivo mPFC-LHA stimulation further increases fat intake after stress, by preventing a decline of interest in the fat. A** Experimental strategy to stimulate mPFC-LHA pathway in control and stressed conditions. **B** Histological verification of mPFC axonal processes in LHA. **C** Bar graphs showing fat intake over the 2 h for experimental groups (Control $n = 27$, Stress $n = 23$ Stress; Two-way ANOVA, main effect treatment, $F_{(1,46)} = 15$, $p = 0.0003$). Main effect stimulation $F_{(1,46)} = 10.38$, $p = 0.002$. Stress plus mPFC-LHA stimulation resulted in the largest consumption of fat (Sidák's multiple comparison tests: Stress−5Hz vs Stress-Mock, $t_{(46)} = 3.27$, $p = 0.01$; Stress−5Hz vs Ctrl−5Hz, $t_{(46)} = 3.78$, $p = 0.003$). **D** Cumulative plots for feeding bouts over the course of two hours for mock and 5 Hz stimulation (Mock Stim: Control $n = 14$, Stress $n = 12$; 5 Hz Stim: Control $n = 13$, 5 Hz $n = 11$). **E** Bar graph quantification of feeding bouts for non-stimulated mice during the Binge and Post-binge phase for controls ($n = 12$) and stressed mice ($n = 14$). Binge phase (0-20 min): unpaired t-test, $t_{(24)} = 3.49$, $p = 0.002$. Post-binge phase (20-120 min):

Mann-Whitney test, $U_{(24)} = 15$, $p = 0.0001$. **F** Bar graph quantification of 5 Hz stimulation effect on feeding bouts during the Binge phase (Control Mock $n = 14$, Control 5 Hz n = 13, Stress Mock $n = 12$, Stress 5 Hz $n = 11$; Two-Way ANOVA, interaction Stimulation- Treatment, $F_{(1,46)} = 0.52$, $p = 0.48$), and the Post-Binge phase (Control Mock $n = 14$, Control 5 Hz $n = 13$, Stress Mock $n = 12$, Stress 5 Hz $n = 11$, Two-Way ANOVA, Stimulation-Treatment interaction, $F_{(1,46)} = 4.05$, $p = 0.05$, Sidák's multiple comparison test mock stim vs 5 Hz in stressed condition, $t_{(46)} = 4$, $p = 0.0004$). **G** Bar graph of data in (**F**) normalized to total feeding bouts indicating persistence of fat consumption beyond the Binge phase (Two-Way ANOVA, interaction Stimulation-Treatment, $F_{(1,46)} = 9.47$, $p = 0.004$, Sidák's multiple comparison test Mock stim vs 5 Hz in stressed condition, $t_{(46)} = 4.825$, $p = 0.0001$). In bar graphs (**C**, **E**–**G**) the superimposed points represent individual animals and error bars represent their SEMs. *$p < 0.05$; **$p < 0.01$; ***=p $< 0.001$.

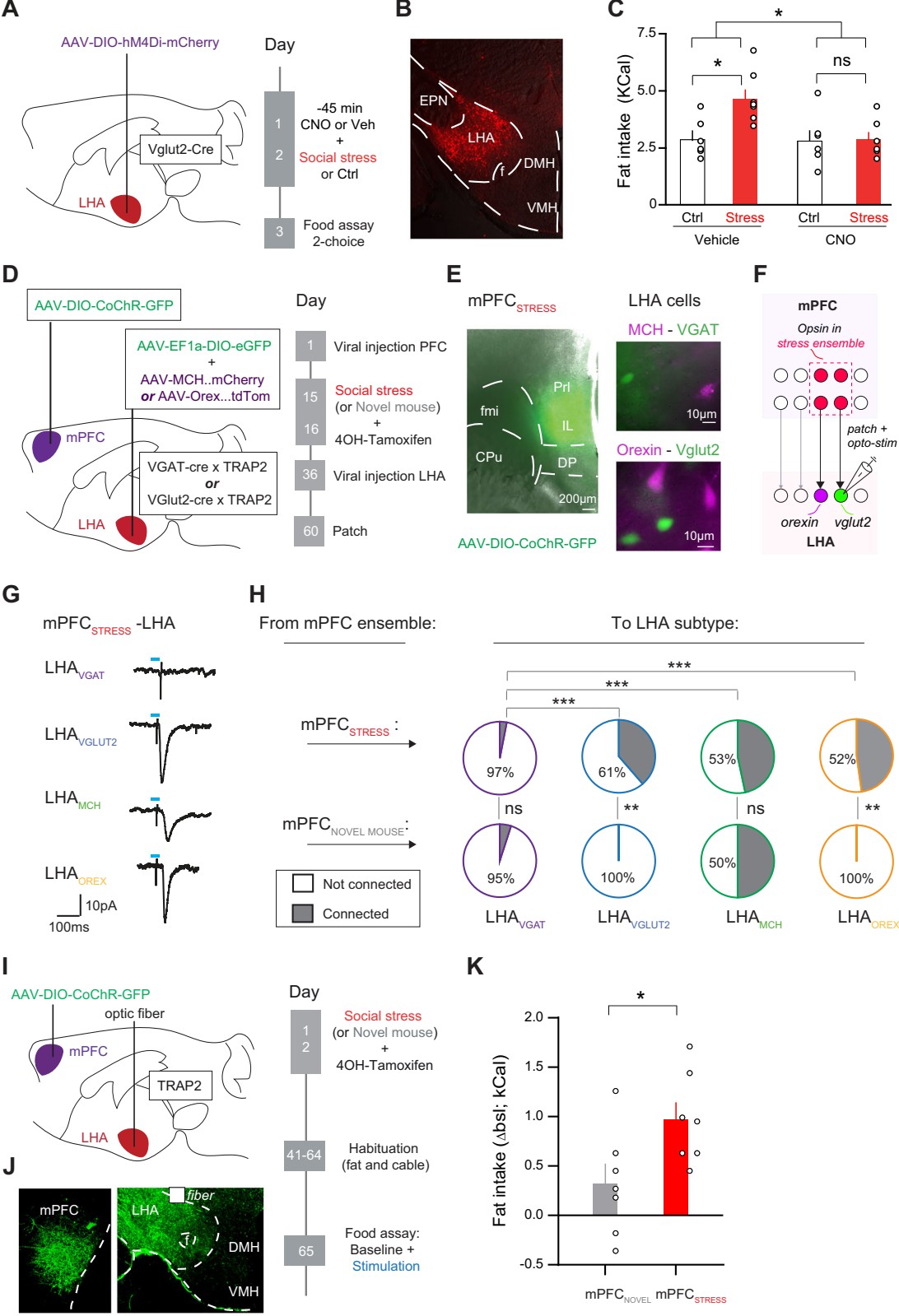

## A role for glutamatergic LHA neurons in transferring effects of stress into binge of fat

We further sought to investigate the importance of LHA neuronal subsets downstream of the mPFC in mediating stress-driven feeding. Glutamatergic LHA neurons are an interesting candidate, as they show stress-driven plasticity of their mPFC synaptic inputs (Fig. 5A, B). Moreover, LHA$_{VGLUT2}$ neurons show a strong increase in calcium

reactivity during social stress[3]. We first asked whether inhibiting the activity of LHA$_{VGLUT2}$ neurons during social stress would prevent subsequent binge eating. To test this, we injected a Cre-dependent inhibitory DREADD (AAV-DIO-hM4Di-mcherry) in the LHA of Vglut2-Cre mice (Fig. 7A, B). We subsequently subjected the mice to either the two-day social stress paradigm or its control. Prior to such stress (or control) events, we administered either vehicle or CNO. Chow intake in

**Fig. 7 | LHA glutamatergic neurons are important mediators of the effects of stress on food intake. A** Schematic for viral targeting and experimental timeline. **B** Histological verification of targeting LHA with chemogenetic actuator (red). **C** Bar graph of fat intake over 2 h (Vehicle-control $n = 6$, Vehicle-stress $n = 7$, CNO-control $n = 6$, CNO-stress $n = 6$; Two Way ANOVA, Stress-CNO interaction $F_{(1,21)} = 4.408$, $p = 0.048$. Sidak post hoc comparisons vehicle-control versus vehicle-stress, $p = 0.034$; CNO-control versus CNO-stress, $p = 0.99$; Vehicle-stress versus CNO-stress, $p = 0.03$. **D** Schematic and timeline for TRAPing mPFC ensemble during social stress (mPFC$_{SOCIAL\ STRESS}$) or during novel mouse exposure (mPFC$_{NOVEL\ MOUSE}$). **E** Left: Representative image of opsin expression after TRAP recombination after an acute stress episode in mPFC. Scale bar 200 μm. Right: Representative images of viral fluorophore expression for co-labeling distinct LHA populations. Top: MCH (red) and VGAT (green) neurons, Bottom: Orexin (red) and Vglut2 (green) neurons. **F** Schematic approach of patch recordings from specific LHA neuron types in brain slices, while optogenetically stimulating mPFC ensemble axons. **G** Representative examples of averaged mPFC$_{SOCIAL\ STRESS}$ opto-evoked EPSC in LHA$_{VGAT}$ (purple), LHA$_{VGLUT2}$ (blue), LHA$_{MCH}$ (green), and LHA$_{OREX}$ (yellow) neurons. Blue rectangle indicates 1 ms opto-stimulation. **H** Pie charts for connectivity of mPFC$_{SOCIAL\ STRESS}$ and mPFC$_{NOVEL\ MOUSE}$ ensembles with distinct LHA populations (VGAT, Vglut2, MCH, and Orexin). mPFC$_{SOCIAL\ STRESS}$ ensemble connected more with glutamatergic than GABA LHA subtypes ($n$-VGAT = 31, 6 mice; $n$-Vglut2 = 49, 11 mice; $n$-MCH = 15, 5 mice; $n$-Orexin = 23, 8 mice; Pearson Chi-Square test, Z = 21, $p = 0.0007$). mPFC$_{SOCIAL\ STRESS}$ ensemble connected more to LHA$_{VGLUT2}$ (Pearson Chi-Square test, Z = 7.14, $p = 0.008$); and LHA$_{OREX}$ neurons (Pearson Chi-Square test, Z = 8.37, $p = 0.004$) than did the mPFC$_{NOVEL\ MOUSE}$ ensemble. **I** Schematics for approach and timeline for ensemble pathway stimulation in the 2-choice food assay. **J** Histological validation of the opsin in mPFC ensemble somata (left) and axons in LHA (right). **K** Bar graph quantification of fat intake over the 2 h for mPFC$_{SOCIAL\ STRESS}$ ($n = 7$) and mPFC$_{NOVEL\ MOUSE}$ ($n = 7$) ensemble projection to LHA, as a delta from baseline. Mann-Whitney U = 7.5, $p = 0.027$. In bar graphs (**C**, **K**) the superimposed points represent individual animals and error bars represent their SEMs. $*p < 0.05$; $**p < 0.01$; $***p < 0.001$.

general was minimal and unaffected (not shown). Importantly, we observed that inhibition of LHA$_{VGLUT2}$ neurons during social stress episodes fully blocked subsequent binge eating of fat (Fig. 7C), indicating that these neurons are important for the induction of stress-eating.

We next evaluated whether the activity of LHA$_{VGLUT2}$ neurons could also contribute to the effects of optogenetic stimulation of the mPFC-LHA pathway on food intake. In Vglut2-Cre mice, we injected AAV-hSyn-CoChR-GFP in the mPFC and placed optic fibers in the LHA for pathway stimulation. We also injected an AAV-DIO-hM4Di-mCherry in the LHA to be able to chemogenetically inhibit LHA$_{VGLUT2}$ neurons (Supplementary Fig. 7B, C). We then assessed food intake during a baseline period in the 2-choice paradigm (Fig. 1C). After baseline periods, in randomized order, we re-exposed mice to the 2-choice paradigm during one of four conditions: optogenetic stimulation of the mPFC-LHA pathway or not, with chemogenetic inhibition of LHA$_{VGLUT2}$ neurons (CNO administration prior to the test) or not. When we did not inhibit LHA$_{VGLUT2}$ neurons, we again observed an increase in fat intake in response to pathway stimulation (Supplementary Fig. 7D). Chemogenetic inhibition of LHA$_{VGLUT2}$ neurons (without stimulation) also gave a trend towards increased food intake. If the sole role of LHA$_{VGLUT2}$ neurons were to inhibit food intake, then mPFC-LHA stimulation without LHA$_{VGLUT2}$ contributions (i.e., mainly favoring mPFC-innervated LHA$_{VGAT}$ recruitment) should make mPFC-LHA stimulation even more potent in driving food intake. Instead, we observed that optogenetic stimulation of the mPFC-LHA pathway during LHA$_{VGLUT2}$ inhibition did not significantly further increase the intake of fat beyond LHA$_{VGLUT2}$ inhibition itself (Supplementary Fig. 7D). This did not appear to be due to a time-constraint limit on feeding, as overnight food-deprived animals still consumed considerably more in the same time period (Supplementary Fig. 7D).

Together, these results suggest that activity of LHA$_{VGLUT2}$ neurons during stress is required for subsequent excess food intake, and it also suggests that despite many LHA$_{VGLUT2}$ neurons curtailing food intake, the activity of at least some LHA$_{VGLUT2}$ subsets also participates in the mPFC-LHA pathway ability to drive food intake upon stimulation.

## Stress-sensitive mPFC neurons project to glutamatergic LHA neurons and can drive increased intake of fat

We then employed another strategy to try to parse out the most relevant branches in the mPFC-LHA network involved in stress-driven food intake. For this, we used an activity-dependent genetic labeling approach and investigated the neurons in the mPFC-LHA axis sensitive to stress, to determine their role in feeding. We expressed an opto-genetic construct in those mPFC cells that were reactive during the social stress (as opposed to all mPFC neurons). For this, we used TRAP2 (Targeted Recombination in Active Populations) transgenic mice, in which the immediate early gene c-Fos can drive Cre-ERT2 expression,

which can then be conditionally transported from the cytosol to the nucleus in a manner dependent on 4-OH tamoxifen[36]. We previously showed that this strategy can be leveraged to capture social stress-reactive neuronal ensembles[37]. We crossed these TRAP2 transgenic mice with either VGAT-Cre or VGLUT2-Cre mice and used the resultant offspring. We first injected an AAV driving Cre-dependent CoChR in the mPFC and exposed the mice to two stress episodes, where each episode was paired with a 4-OH tamoxifen injection. Weeks later, we also injected AAVs in the LHA to fluorescently label specific sub-populations such as LHA$_{VGLUT2}$, LHA$_{VGAT}$, LHA$_{OREX}$, and LHA$_{MCH}$ cells as before (Fig. 7D–F; Supplementary Fig. 7B, C). Importantly, while both TRAP2 transgenic mice and VGAT-Cre/VGLUT2-Cre transgenic mice rely on conditional expression via the same Cre-LoxP system, the viral injection scheme in this case avoided interference between them. First, we avoided mPFC CoChR expression in a manner not linked to the TRAPing of an ensemble in the VGLUT2-Cre mice, on the basis of mPFC neurons expressing VGLUT1 rather than VGLUT2[21,38] (Supplementary Fig. 7E). We also avoided that there would be Cre-dependent fluorophore recombination in the LHA driven by the TRAP system rather than on the basis of LHA cell type identity, by performing the LHA viral injection several weeks after the TRAPing event in the mPFC had taken place (Fig. 7D), well outside the time window of 4OH-tamoxifen dependent Cre-driven recombination[36,37]. Finally, we also took along a control ensemble of exposure to a novel mouse rather than stressful altercations (Fig. 7D). In both the social stress TRAP experiment and the novel mouse TRAP experiment, we observed expression of CoChR in mPFC ensembles (Fig. 7E; Supplementary Fig. 7F). We then performed patch-clamp recordings in the LHA and assessed whether optogenetic stimulation of mPFC$_{NOVEL\ MOUSE}$ or mPFC$_{SOCIAL\ STRESS}$ ensembles resulted in synaptic contacts with specific LHA cell types. Stimulating the mPFC$_{NOVEL\ MOUSE}$ ensemble caused relatively little connectivity with the LHA, though there was connectivity with LHA$_{MCH}$ neurons (Fig. 7G, H). In contrast, optogenetically stimulating the mPFC$_{SOCIAL\ STRESS}$ ensemble resulted in synaptic responses in LHA$_{VGLUT2}$, LHA$_{MCH}$, and LHA$_{OREX}$ neuronal subsets (Fig. 7G, H). Instead, LHA$_{VGAT}$ neurons received almost no input (Fig. 7G, H).

These results suggest that mPFC neurons that are modulated within the time window of several hours during social stress experiences send preferential projections to glutamatergic neurons (including their peptidergic subtypes[18]) over GABAergic LHA neurons. Given that we had shown the importance of glutamatergic LHA neurons in mediating the effects of stress on food intake (Fig. 7C), we next asked whether optogenetically stimulating mPFC$_{SOCIAL\ STRESS}$ projections to these glutamatergic LHA neurons would modulate food intake. We used TRAP2 mice in which we again expressed an opto-genetic construct (AAV-DIO-CoChR-GFP) in the mPFC, placed an optic fiber above the LHA, and triggered recombination of the opsin in the mPFC either by pairing the social stress experiences or the novel

mouse experiences with 4OH-tamoxifen injections (Fig. 7I, J). After recovery, we optogenetically stimulated these pathways with 5 Hz in the context of the 2-choice food assay. We observed that optogenetic stimulation of the mPFC$_{SOCIAL\ STRESS}$ to LHA pathway resulted in increased fat intake (Fig. 7K), without altering chow intake (Supplementary Fig. 7G). Together, these results suggest that glutamatergic LHA neurons are important downstream targets of the mPFC for the effects of stress on increased food intake.

## Discussion

Uncovering the neurobiological substrate for stress-driven overconsumption of palatable foods is an important topic, also in view of the contributions of such behavior to eating disorders like Binge Eating Disorder and Bulimia Nervosa, and to obesity[2,39]. The prefrontal cortex is implicated in binge eating disorders[9], but it has been unclear through which network adaptations it could orchestrate stress-driven food intake. Here we show that there is direct mPFC control over distinct LHA circuits, and that stress-driven alterations of the mPFC-LHA network contribute to overconsumption of palatable fat. Here, we discuss multiple facets of these findings.

### The mPFC-LHA pathway has a conditional role in food intake, driving excess intake

Our pathway stimulation and inhibition experiments reveal that the mPFC-LHA network plays a conditional role in food intake. First, the role of the pathway in feeding was only unmasked by stress. Chemogenetic inhibition of the pathway under normal (non-stressed) conditions did not alter food intake, in accordance with an earlier study[21]. Instead, we show that after a social stress experience that drives excess intake of specifically palatable food[3], inhibition of the pathway abolishes not all feeding, but rather only the amount of excess fat intake due to stress. This suggests that the pathway is not generally necessary for feeding behavior, but its importance emerges under certain conditions, such as a stressed state.

Second, mPFC-LHA pathway stimulation increased the intake of palatable fat, without affecting concurrently available chow. This indicates that the circuit manipulations caused a directed increase in drive towards palatable foods, rather than an indiscriminate urge to chew. Although not explicitly tested here, we consider it likely that the stress-driven effects mediated by the mPFC-LHA network result in an increased preference for palatable food in general, rather than a nutrient-specific preference for fat. In support of this, moderate social stress causes preference for palatable food in general, including both fat and sucrose[3]. Third, our data suggest that internal states may contribute to the role of the mPFC-LHA pathway in food intake. We observed that the optogenetic stimulation of the pathway did not cause altered food intake at the start of feeding sessions, when the drive for food intake was still high. Instead, it was only at later timepoints, when normally food interest would have waned, that pathway stimulation caused continued food interest and time spent eating. This finding also indicates that the role of mPFC-LHA networks will likely depend on whether or not animals are food-deprived. In our pathway manipulation experiments, animals were ad libitum-fed, perhaps allowing disinterest in food to occur more quickly under normal circumstances, and revealing a role for mPFC-LHA network activity more readily.

Our fiber photometric data show that the mPFC-LHA pathway shows a transient decrease in activity during food intake, which reverses upon the end of contact with the food, in accordance with a previous study[21]. It is of note that increasing mPFC-LHA pathway activity, at least at certain frequencies, drives food intake, whereas mPFC-LHA pathway activity decreases during the actual food intake. It is interesting to compare this with other systems with a link to feeding. For instance, arcuate nucleus AgRP neurons are potent drivers of food intake when stimulated yet also show dips in their activity during food

intake. This may reflect an encoded energy need signal of these neurons that is then attenuated during caloric intake[40–43]. Somewhat analogously, mPFC-LHA pathway activity may reflect not a generic energy deficit signal, but more a conditional one, promoting further intake of palatable energy-dense foods under specific circumstances, such as after stress. Another interpretation is that mPFC-LHA pathway activity reflects an engagement signal to continue to seek out palatable food instead of moving on to other activities. Actual consumption of the food may then require a brief decrease in such activity to allow for consumption instead of seeking/approach behaviors. Recent evidence does indeed suggest that cortical inputs to the LHA participate in the encoding of behavioral transitions, including those linked to feeding[44].

Overall, these data suggest that the mPFC-LHA network activity, while not indispensable for regular food intake, does become critically important for excessive intake during stress-driven hyperphagic states. This may be due to pathway activity indicating further energy needs, overriding satiety states, and/or by causing continued food engagement to facilitate excessive intake in short periods of time.

### The mPFC-LHA is not a singular pathway, but a multi-branched network

Rather than a singular pathway, we now show that the mPFC-LHA network is composed of multiple branches that can have diverse functions. mPFC neurons innervate multiple LHA subpopulations based on LHA cell type. We observed strong connectivity with LHA$_{VGAT}$ and LHA$_{VGLUT2}$ neurons, and with LHA$_{OREX}$ and LHA$_{MCH}$ neurons, which, according to transcriptomic analyses, represent glutamatergic VGLUT2-positive subsets[18]. In the current study, we also identified that mPFC-innervated glutamatergic LHA neurons project to multiple downstream targets, including VTA, LHA, and Peri-PVN. Within the current study, we did not further investigate distinct projection targets of mPFC-innervated LHA GABAergic or specific neuropeptidergic populations, but it is likely that these too will have multiple distinct targets[15]. Global mPFC-LHA pathway manipulation or measurement studies, need to therefore consider that the effects observed represent the net total of multiple sub-pathways, which likely can be in part functionally distinct from each other.

To what extent are these branches within the mPFC-LHA network anatomically distinct from each other, as opposed to representing collaterals? At the LHA level, prior evidence does suggest that, for instance, LHA glutamatergic neurons with distinct output targets are at least distinct subsets. Such as LHA$_{VGLUT2}$-VTA and LHA$_{VGLUT2}$-LHb neurons, which are anatomically and molecularly distinct sets[28]. At the level of the mPFC, it is an open question to what extent different or the same mPFC neurons innervate distinct LHA neuronal subsets (e.g., LHA$_{GABA}$ versus LHA$_{GLUTAMATE}$). However, our ensemble-based tagging experiments do indicate that there is at least no full collateralization. We observed that there are mPFC neurons that, when stimulated, do synapse onto glutamatergic but not on GABAergic LHA neurons, or that even synapse on specific neuropeptidergic populations such as LHA$_{MCH}$ neurons.

Further studies will need to unravel which distinguishable mPFC neurons are embedded in different anatomical and functional branches within the larger mPFC-LHA network.

### The mPFC projections to LHA and its glutamatergic neurons as a (social) stress integrator

In this study, we captured single-cell dynamics of mPFC cells projecting to the LHA during social stress experiences with in vivo electrophysiology. We observed a heterogeneous response of these cells, with some excited and some inhibited. Bulk photometric calcium recordings reflected a biphasic population level response, which on the basis of our in vivo electrophysiology data, we consider to be likely mediated by separate subsets of neurons. There was a modest enrichment of stress-modulated responses within the mPFC-LHA pathway

compared to non-optotagged mPFC neurons, in accordance with findings that other mPFC pathways are also stress-sensitive[7]. Social stress is a naturalistic but complex stimulus (e.g., featuring stress, social interaction, pain, movement). Social interaction in itself can modulate mPFC-LHA[20,31,42], but it is likely that the stressful aspects of the stimulus are also engaging the network. Our patch clamp data also showed that restraint stress, which has very different features from social stress as an experience (e.g., non-social vs social, immobilization vs bursts of movement), caused increased food intake and similar synaptic alterations at mPFC-LHA$_{VGLUT2}$ synapses as did social stress. Also, a previous study has shown that the mPFC-LHA pathway is sensitive to another type of aversive stimulus, namely air puff stimuli[21].

Our in vivo electrophysiology studies show that during stress, some neurons exhibit an increase in activity and others a decrease. Given the multiple types of connections made downstream in the LHA, it is possible that these different stress-induced responses in mPFC neurons correspond to distinct branches of the mPFC-LHA network. There is evidence to suggest that at least mPFC neurons projecting to LHA glutamatergic neurons are amongst those that are excited by social stress. Our previous work showed that glutamatergic LHA neurons themselves show an immediate strong excitation during social stress[3], and we here show that a high percentage of these neurons (~80%) receive excitatory input from the mPFC. Furthermore, in the current study, we show that activity in these LHA glutamatergic neurons during social stress is indispensable for the subsequent hyperphagia. We also demonstrate that optogenetically stimulating stress-recruited mPFC ensemble projections to glutamatergic LHA subsets can drive feeding. Overall, these findings point to mPFC control over LHA glutamatergic neuronal subsets as a key substrate in driving stress-driven feeding.

### Stress-driven plasticity within mPFC-LHA circuits

After social stress, at a time point where mice would overeat[3], we observed synaptic changes at mPFC-LHA$_{VGLUT2}$ synapses. Specifically, we observed evidence for weaker mPFC glutamatergic inputs onto the average LHA$_{VGLUT2}$ cell, via reduced release probability at mPFC presynaptic elements. Interestingly, many LHA$_{VGLUT2}$ cells are linked to a role in curtailing food intake[15,17]. Consequently, a reduction of the excitatory input on such neurons could be in accordance with a weakening of a brake on the drive for food intake. A finer-grained analysis revealed that the effect of stress on mPFC synapses onto LHA$_{VGLUT2}$ cells was further dependent on the projection targets of the hypothalamic cells. We observed evidence for weakened glutamatergic mPFC synapses onto those LHA$_{VGLUT2}$ cells projecting to a midline hypothalamic region, which, upon stimulation, curtails food intake[45]. Oppositely, we found that glutamatergic mPFC synapses onto LHA$_{VGLUT2}$ cells projecting to the VTA exhibit potentiation after stress. This latter finding is in accordance with our recent report that, after social stress, LHA$_{VGLUT2}$ projections to VTA dopamine neurons become potentiated and that this plays a causal role in excess food intake[3]. LHA$_{VGLUT2}$ cells have several more projection targets than the ones investigated here[46], which may also be changed by stress. Therefore, the analysis of these specific projections is not meant to be exhaustive, but it illustrates that mPFC synaptic contacts onto distinct LHA$_{VGLUT2}$ subpopulations are divergently affected by social stress.

Aside from mPFC-LHA branches that did exhibit stress-driven plasticity, there were also several that did not (e.g., mPFC synapses onto LHA$_{VGAT}$ neurons). Of note is that even if mPFC synapses onto LHA$_{VGAT}$ neurons did not, on average, show stress-driven plasticity, these connections are still engaged when stimulating the mPFC-LHA pathway, potentially contributing to increasing food intake. The findings of plasticity in certain branches should then also be observed in the context of the network as a whole. We do not claim here that one particular sub-branch is solely responsible for stress-driven feeding. Rather, stress has multiple actions in the network. mPFC-LHA branches

(conditionally) linked to promoting food intake are either strengthened or unaffected by the stress experience, whereas branches linked to promoting food intake are either weakened or unaffected. As a whole, this likely shifts the mPFC-LHA network more towards favoring further intake of palatable food.

One relevant aspect that our study did not address is the extent to which individual animals may be more prone to exhibiting stress-driven overconsumption of palatable food, and whether this depends on individual differences in alterations in the mPFC-LHA pathway. Chronic social stress has been shown to have heterogeneous effects on mice in terms of willingness to engage in social interactions and extent of induced anhedonia. Classification systems of (distinct types) of stress-susceptible and unsusceptible animals[47,48] have been informative in this regard. Future studies will need to address whether moderate social stress affects increased palatable food intake, and/or related eventual weight gain over time[23] also clustering in distinct behavioral profiles.

### Different types of mPFC-LHA stimulation result in different outcomes regarding food intake

Prior studies showed that chemogenetic (i.e., likely strong and continuous) or high-frequency optogenetic stimulation of the mPFC-LHA pathway reduced intake of chow in mice[21]. Here we show with our global pathway manipulations that optogenetic stimulation of the mPFC-LHA network, at lower frequencies, can also increase food intake. We found that continuous periods of 5 Hz stimulation drove food intake, whereas continuous periods of 1 Hz or 10 Hz stimulation did not. Interestingly, a previous study reported that 5 Hz optogenetic stimulation of LHA projections to the VTA could more potently drive food intake than either 2.5 Hz or 20Hz[19]. The lack of effect on food intake of mPFC-LHA with 1 Hz stimulation may have been due to insufficient engagement of the mPFC-LHA network. Continuous 10 Hz stimulation was qualitatively distinct from 5 Hz stimulation in terms of its effect on food intake. This could not be explained by an ineffective stimulation in itself, as animals exhibited more locomotor activity to the 10 Hz stimulation, which in itself might have contributed to the lack of increased intake of food as compared to stimulating the pathway with 5 Hz. At the mechanistic level, our ex vivo data showed, that over time, 10 Hz stimulation of mPFC inputs was less easily followed by LHA$_{VGAT}$ neurons than by LHA$_{VGLUT2}$ neurons in terms of synaptic transmission fidelity. It is therefore possible that the lack of a 10 Hz effect was more due to the continuous nature of delivery rather than the frequency per se. These data, together with previous reports, show that distinct ways of mPFC-LHA stimulation can produce differential effects on feeding behaviors, possibly by striking a different balance in the sustained engagement of distinct mPFC-LHA subbranches.

Overall, our work identifies the mPFC-LHA as a multi-branched network, with an important role in particular for mPFC control over glutamatergic LHA neuronal populations with regard to integrating information about stress and the subsequent excess intake of palatable food sources such as fat. These findings may ultimately be relevant in the context of eating disorders, in which stress triggers bouts of binge eating behavior.

## Method
### Animals
In all experiments, naïve adult male mice were used (20-35 g, >6 weeks). C57Bl6J (Jax #664), Vglut2-Cre[49] heterozygous (Jax #28863), VGAT-Cre[49] heterozygous (Jax #016962), and TRAP2 heterozygous[36] (Jax #030323), animals were bred in-house but originated from the Jackson Laboratory. Swiss-CD1 mice (35–45 g, >12 weeks), were purchased from Janvier (France). All mice (except Swiss-CD1 mice) were group-housed (2–5 per cage) unless otherwise specified. Mice were housed in a 12:12 light/dark cycle (lights on at 07:00 a.m.) at 22 ± 2 °C (60–65% humidity). Unless otherwise specified (during brief binge

eating recording sessions, see below), animals had access to ad libitum water and lab chow (3.61 kcal/g, CRM (E), 801730, Special Diets Services). Experiments were approved by the local animal welfare body (AWB, or IvD in Dutch) of Utrecht University/ University Medical Center Utrecht, and by the national authority (CCD). Experiments were conducted in agreement with the Dutch law (Wet op de Dierproeven, 2014) and the European regulations (Guideline 86/609/EEC).

## Stereotaxic surgery

For stereotactic surgery, all mice were anesthetized with ketamine (75 mg/kg i.p.; Narketan, Vetoquinol) and dexmedetomidine (1 mg/kg i.p.; dexdomitor, Vetoquinol). Lidocaine (0.1 ml; 10% in saline; B. Braun) was injected under the skin on the skull a few minutes before surgery. Eye ointment cream (CAF, Ceva Sante Animale B.V.) was applied before surgery. All mice were at least 6 weeks of age at the time of surgery. Animals were fixed on a stereotactic frame (UNO B.V.) and kept on a heat pad during surgery (33 °C). When animals were implanted with either optic fibers or cannulas, the skull surface was scratched with a scalpel, and phosphoric acid (167-CE, ultra-Etch, ultradent, USA) was applied for 5 min to roughen the surface at the start of surgery. Injections were done using a 31 G metal needle (Coopers Needleworks) attached to a 10 μl Hamilton syringe (model 801RN) via flexible tubing (PE10, 0.28 mm ID, 0.61 mm OD, Portex). The Hamilton syringe was controlled by an automated pump (UNO B.V. -model 220). Injections were done bilaterally unless otherwise specified, and volumes ranged between 250 and 300 nl per side, at an injection rate of 100 nl/min. The injection needle was retracted 100 μm, 9 minutes after the infusion, and withdrawn completely 10 minutes after the infusion. The skin was sutured post-surgery (V926H, 6/0, VICRYL, Ethicon). Animals were then subcutaneously injected with the dexmedetomidine antagonist atipamezole (50 mg/kg; Atipam, Dechra), carprofen (5 mg/kg, Carporal), and 1 ml of saline and left to recover on a heat plate of 36 °C. Carprofen (0.025 mg/L) was provided in the drinking water for one week post-surgery. Animals were single-housed post-surgery and rejoined with their cage mates three days after surgery, unless otherwise stated.

## For ex vivo optogenetic-assisted electrophysiology experiments

For transgenic lines: VGAT-Cre and Vglut2-Cre, mice were injected with AAV5-Syn-CoChR-GFP (3.8*10^12 gc/ml; 300 nl; UNC Vector Core) in mPFC (+1.8 mm anterior to bregma; 0.90 mm lateral; -2.2 mm ventral from skull under an angle of 10°) and injected with AAV5-hSyn-DIO-mCherry (3*10^12 gc/ml; 300 nl; Addgene) in LHA (-1.3 mm posterior to bregma; 1.9 mm lateral; −5.4 mm ventral from skull under an angle of 10°) to visualize GABAergic neurons in VGAT-Cre mice and glutamatergic neurons in Vglut2-Cre mice.

To visualize neuropeptide populations in LHA, we used C57bl6 mice, which were injected with the same protocol for the mPFC as the Vglut2-Cre and VGAT-Cre. Mice were injected with rAAV5-MCHpr-Gq-mCherry (1*10^12 gc/ml) for labelling the MCH populations and with AVV5-hORXpr1-3TdTomato (1*10^12 gc/ml) for the Orexinergic neurons. Both were injected in the LHA with the same coordinates and volume as for the transgenic mice.

To identify glutamatergic LHA neurons projecting to LHb, VTA, and Peri-PVN, we used a Cre-dependent retrograde tracer retroAAV-hSyn-DIO-mCherry (2.8*10^12 gc/ml; Addgene) in Vglut2-Cre mice. Mice were then injected in one of these areas: in the LHb (-1.6 mm posterior to bregma; 1.2 mm lateral; -3.05 mm ventral from skull under an angle of 15°), in VTA (-3.2 mm posterior to bregma; 1.65 mm lateral; -5.0 mm ventral from skull under an angle of 15°) and Peri-PVN (-0.8 mm posterior to bregma; 1.1 mm lateral; -5.4 mm ventral from skull under an angle of 10°). Mice were also injected in the mPFC with AAV5-Syn-CoChR-GFP.

**For ex vivo TRAP-directed electrophysiology experiments.** VGAT-cre::TRAP2 and Vglut2-cre::TRAP2 mice were first injected in mPFC with rAAV5-Syn-Flex-CoChR-GFP (4.4*10^12 gc/ml; UNC Vector Core). After a minimum of two weeks, mice were subjected to stress and tamoxifen i.p injection (25 mg/kg). After a minimum of one week, mice were injected in LHA with AAV5-hSyn-DIO-eGFP (1*10^12vc/ml; Addgene) and co-injected with rAAV5-ORXpr1-3TdTomato (1*10^12 gc/ml) or rAAV5-MCHpr-Gq-IRES-mCherry (1*10^12 gc/ml). To manipulate the mPFC ensemble-LHA, TRAP2 mice were bilaterally also injected with rAAV5-Syn-Flex-CoChR-GFP (4.4*10^12 gc/ml; UNC Vector Core, mPFC: +1.8 mm anterior to bregma; 0.90 mm lateral; −2.2 mm ventral from skull under an angle of 10°) and optic fibers (diameter 200 μm; FT200UMT, 0.39NA, Thorlabs, Germany) fitted with ceramic ferrules (230 μm bore size, Precision Fiber Products, MM-FER2002, USA or Thorlabs, CFLC230-10, Germany) were implanted bilaterally in the LHA (-1.2 mm posterior to bregma; 1.85 mm lateral; -4.85 mm ventral from skull under an angle of 10°). After surgery, mice were housed collectively for at least 6 weeks.

**For in vivo optogenetic experiments.** To drive the mPFC-LHA pathway at various frequencies, C57BL6 mice >3 months of age were bilaterally injected with the light-sensitive Channelrhodopsin variant CoChR (300 nl rAAV5-Syn-CoChR-GFP at a titer of 3.8*10^12 gc/ml) or a fluorophore for control animals (300 nl rAAV5-Syn-eYFP, titer of 5.6*10^12 gc/ml) (Penn Vector Core, USA) in the mPFC (AP + 1.8; ML ± 0.92 mm from bregma; DV −2.24 mm from the skull at a 10° angle; 100 nl/min). After injection, a 10-min delay was implemented, and two minutes before retraction, the needles were lifted 100 micrometers to allow for proper diffusion of the virus. To stimulate mPFC axon terminals innervating the LHA, optic fibers (diameter 200 μm; FT200UMT, 0.39NA, Thorlabs, Germany) fitted with ceramic ferrules (230 μm bore size, Precision Fiber Products, MM-FER2002, USA or Thorlabs, CFLC230-10, Germany) were targeted to the lateral hypothalamus and implanted bilaterally at AP −1.2; ML ± 1.85 from bregma and DV −4.85 mm from the skull under a 10° angle. The implants were fixed to the skull using 2 stainless steel screws, C&B-Metabond (Parkell Inc., USA), and dental cement (GC Corporation, Japan). After surgery, mice were individually housed for the remainder of the experiment.

For the optogenetic stimulation of the LHA$_{VGLUT2}$-periPVN pathway, the optogenetic procedure of surgery remained the same, vglut2-cre mice were injected bilaterally with the light-sensitive Channelrhodopsin variant CoChR (300 nl rAAV5-Syn-Flex-CoChR-GFP 2.2*10^12 gc/ml, UNC Vector Core) or with the fluorophore for control (300 nl rAAV5-Syn-DIO-GFP 2.2*10^12 gc/ml, Addgene) in LHA (−1.2 mm posterior to bregma; 1.85 mm lateral; −4.85 mm ventral from skull under an angle of 10°), and the optic fibers were place above the peri-PVN (−0.8 mm posterior to bregma; 1.1 mm lateral; -4.85 mm ventral from skull under an angle of 10°).

After surgery, Vglut2-cre mice were housed collectively for at least 8 weeks for the PFC-LHA$_{VGLUT2}$ experiments and a minimum of 6 weeks for LHA$_{VGLUT2}$-periPVN.

**For in vivo chemogenetic experiments.** The pAAV-hSyn-DIO-hM4D(Gi)-mCherry (DREADD virus) (4*10^12 gc/ml, Addgene, United States, addgene.org) or pAAV-hSyn-DIO-mCherry (4.2*10^12 gc/ml, Addgene, United States, addgene.org) were bilaterally injected into the mPFC (in mm: +1.8AP, −1.2 ML from Bregma and, −2.3 DV, from skull, at a 10° angle) with an injection-rate of 100 nl/min, and a volume of 300 nl. The Cav2-cre-virus (2*10^12 gc/ml, Plateforme de Vectorologie de Montpellier, https://plateau-igmm.pvm.cnrs.fr/), combined with red-retrobeads (60x dilution, Lumafluor Inc, lumafluor.com) for injection site verification, was bilaterally injected into the anterior LHA (in mm: −1AP, 1.85 ML, 5.36DV from Bregma, at a 10° angle), and posterior LHA (in mm: −1.85AP, 1.9 ML, 5.41DV, from Bregma, at a 10° angle) with an injection-rate of 100 nl/min. After injection of the viral

construct, the needle was kept in place for 8 min after which it was retracted for 100 um and left in place for two more minutes to allow proper diffusion of the virus. Thereafter, the needle was fully subtracted at a slow pace. After injection, the skin was sutured (V926H, 6/0, VICRYL, Ethicon). After surgery, mice were individually housed for the remainder of the experiment.

For experiments where we optogenetically stimulated mPFC-LHA while having chemogenetic control over LHA$_{VGLUT2}$ neurons, we used VGLUT2-cre mice. The pAAV-hSyn-DIO-hM4D(Gi)-mCherry DREADD virus (300 nl 4*10^12 gc/ml, Addgene, United States) was injected in the LHA and to combined with the optogenetic optic fibers were implanted bilaterally in the LHA (−1.2 mm posterior to bregma; 1.85 mm lateral; -4.85 mm ventral from skull under an angle of 10°) and the light-sensitive Channelrhodopsin variant CoChR (300 nl rAAV5-Syn-CoChR-GFP at a titer of 3.8*10^12 gc/ml, UNC Vector Core) was injected bilaterally in the mPFC (+1.8 mm anterior to bregma; 0.90 mm lateral; −2.2 mm ventral from skull under an angle of 10°).

**For in vivo electrophysiology with optrode experiments.** To identify mPFC neurons that project to the LHA in electrophysiological recordings, we used a dual viral approach. First the Cav2-cre-virus (2*10^12 gc/ml, Plateforme de Vectorologie de Montpellier, https://plateau-igmm.pvm.cnrs.fr/) was injected, combined with red-retrobeads (60x dilution, Lumafluor Inc, lumafluor.com) for injection site verification, into LHA (in mm: −1.3AP, 0.9 ML, -5.3DV from Bregma, 0° angle). A second virus pAAV-EF1α-doublefloxed-hChR2(H134R)-mCherry was injected at two locations in the mPFC (+1.8 mm AP, -0.45 mm ML, −2.5 mm DV and +1.8 mm AP, −0.45 mm ML, −1.8 mm DV from Bregma, 0° angle, 100–150 nl per site and a titer of 2–4*10^12 gc/ml). All injections were done unilaterally and targeting the left hemisphere, except for 2 mice, which received a bilateral injection in the LHA. Injections for electrophysiological recordings were done using glass pipettes (#504949, World Precision Instruments) coupled to a Nanojet injector (#504127, World Precision Instruments). To allow for proper expression of the virus before electrophysiological recordings, the optrode was placed in a second surgery, >3 weeks after viral injections, in the left mPFC (+1.8 mm AP, −0.5 mm ML from Bregma, −2.0 mm DV from the surface of the brain). During this surgery, a craniotomy was made above the mPFC and the dura removed, whereafter the optrode was inserted in the brain using a micromanipulator (Scientifica, UK). Two screws were placed in the skull for attachment of the optrode to the skull, and the ground was placed through a small hole in the skull on the cerebellum and fixed with a small drop of superglue. After closure of the craniotomy with silicone, the whole implant was fixed with C&B-Metabond (Parkell Inc., USA), and dental cement (GC Corporation, Japan). Recordings during behavior were made at least 3 weeks after implantation of the optrode to allow for proper recovery. After surgery, mice were individually housed for the remainder of the experiment.

**For in vivo fiber photometry experiments.** For in vivo fiber photometry we unilaterally injected a retrograde CAV2-cre virus (2 × 10^12 gc/ml, 200 nl per site) in the LHA (AP −1.0; ML 1.85; DV −5.40 & AP −1.85; ML 1.9; DV −5.40) and rAAV-syn-FLEX-jGCaMP8s-WPRE (3*10^12 gc/m, 300 nl) in the PFC (AP +1.8; ML 0.90; DV −2.2) of C57BL/6 mice. Next, we inserted an optical fiber (ø400 μm, Thorlabs) above the injection site (DV: −1.9). We fixed the fiber with superbond glue (Sun Medical Co.), and dental cement (Fuji PLUS-capsules, G.C. corporation). After the surgery, mice were allowed to recover for 4 weeks, while being socially housed, before the behavioral recordings started to ensure virus expression.

**Patch-clamp Electrophysiology**
After a minimum of six weeks following surgery, mice were anesthetized with isoflurane (Zoetis, UK) in the morning and were then rapidly decapitated. Coronal brain slices of 250 μm were cut on a vibratome (1200 VTs, Leica, Rijswijk, The Netherlands) in ice-cold carbogenated (95% O$_2$, 5% CO$_2$) cutting solution, which contained (in mM) choline chloride (92); ascorbic acid (10); CaCl$_2$ (0.5); glucose (25); HEPES (20); KCl (2.5); N-Acetyl L Cysteine (3.1); NaHCO3 (25); NaH2PO4 (1.2); NMDG (29); MgCl$_2$ (7); sodium pyruvate (3); Thiourea (2). Slices were transferred for 5–10 min to a warmed solution (34 °C) of the same composition, before they were kept at room temperature in carbogenated incubation medium, which contained (in mM): ascorbic acid (3); CaCl$_2$ (2); glucose (25); HEPES (20); KCl (2.5); NaCl (92); NaHCO3 (20); NaH2PO4 (1.2); NMDG (29); MgCl$_2$ (2); sodium pyruvate (3) and Thiourea (2). During recordings, slices were immersed in artificial cerebrospinal fluid (ACSF) containing (in mM): NaCl (124); KCl (2.5); CaCl$_2$ (2.5); glucose (11); HEPES (5); NaHCO3 (26); NaH2PO4 (1); MgCl$_2$ (1.3) and were continuously superfused at a flow rate of 2 ml min$^{-1}$ at 32 °C.

Neurons were patch-clamped using borosilicate glass pipettes (2.7–4 MΩ; glass capillaries, GC150-10, Harvard apparatus, UK), under a BX51WI Olympus microscope (Olympus, France). For voltage or current clamp recordings, the signal was amplified, low-pass filtered at 2.9 kHz with a 4-pole Bessel filter, and digitized at 20 kHz with an EPC9/2 dual patch-clamp amplifier (HEKA Elektronik GmbH). Data were acquired using PatchMaster v2x78.2 software. After the break-in, 10 minutes of waiting time occurred prior to the start of the recording. Series resistance was determined in Voltage clamp with a − 4 mV step from −65 mV to −69 mV lasting 50 ms, using the resultant capacitive current transients and Ohm's law. Light pulses (470 nm, 1 ms; 25.4 mW) were delivered with a pE-4000, CoolLed, UK illumination system. All electrophysiological measures are recorded with a 10 s inter-sweep interval (0.1 Hz), and all data points are shown as an average of 10-20 sweeps, unless otherwise specified.

**Paired-pulse ratio (PPR).** PPR experiments were performed with cesium methanesulfonate-based internal medium, containing (in mM) cesium methanesulfonate (139), CsCl (5), HEPES (10), EGTA (0.2), creatine phosphate (10); Na2ATP (4); Na3GTP (0.3), MgCl2 (2). Averaged AMPAR-mediated currents in LHA neurons were measured in response to optogenetic stimulation of mPFC terminals using two pulses with a 60, 100, or 200 ms inter-pulse interval and measuring the resulting peak amplitude of the two synaptic responses. The PPR was calculated by dividing the amplitude of the synaptic response to the 2nd pulse by the amplitude of the synaptic response to the 1st pulse.

**NMDAR-AMPAR ratios.** Synaptic LHA optogenetically evoked AMPAR and NMDAR currents were measured in response to a single light pulse. Recordings were made in voltage clamp with the aforementioned cesium methanesulfonate-based internal medium. Under these conditions, it was possible to detect inward glutamatergic currents (at −65 mV) and NMDAR (at 40 mV at 100 ms). Experiments were done in the presence of 100 μM picrotoxin, to block potential GABA$_A$ receptor contributions. The NMDA/AMPA ratio was calculating by dividing the amplitude of the EPSC after 100 ms at +40 mV by the amplitude of the synaptic response at -65mV.

**mPFC-LHA connectivity experiments**
Recordings were made in voltage clamp in a cesium methane sulfonate based internal medium as described above. Under these conditions, it was possible to detect inward glutamatergic currents (at −65 mV). Cell type specific connectivity was determined based on whether a neuron showed an optogenetically-evoked synaptic response larger than 5 pA. For monosynaptic confirmation, cells were recorded and optostimulated (1 ms) once per sweeps, first during at least 15 sweeps with ACSF, then 1 μM of TTX was added to ACSF for a minimum of 25 sweeps to observe the effect on the synaptic response and finally 100 μM 4-AP was added to the mix and recorded for a least 20 sweeps.

**Intrinsic excitability and mPFC-driven LHA cell firing experiments.** Recordings were made in current clamp in a potassium gluconate-based internal solution containing (in mM), Potassium Gluconate (139); HEPES (10); EGTA (0.2); creatine phosphate (10); KCl (5); Na2ATP (4); Na3GTP (0.3); MgCl2 (2). Only LHA cells showing a synaptic response larger than 5pA at -65mV after 1 ms of opto-stimulation went under the following protocol. To determine current-voltage relationships and biophysical properties, membrane resistance and firing pattern, cells were recorded in current clamp. Cells were clamped at 0 pA and from there were subjected to 17 subsequent sweeps. Every sweep started with a 0 pA injection of 400 ms, followed by a current step of 800 ms length of −150 pA (with a + 25 pA increasing increment on each subsequent sweep), followed by a return to 0 pA for 1000 ms. Inter-sweep interval was 10 s.

To assess if the opto-stimulation of mPFC inputs was sufficient to fire the LHA neurons, cells were clamped at −50pA to prevent the spontaneous firing, and 25 pulses of 1 ms were applied at 5 Hz for 10 sweeps.

### Spike fidelity over 10 min

Recordings were made in current clamp in a potassium gluconate-based internal (see above), cells were clamped at 0pA, except if they were spontaneously firing (-20pA or -30pA); and were opto-stimulated during 10 min at 1 Hz, 5hz and 10 Hz in a random order. EPSP was counted if the change of amplitude was equal to or more than 2 mV compared to the overall baseline.

**Drugs for patch-clamp experiments.** All drugs were obtained from Sigma (UK), Abcam (UK), and Tocris (UK). With the exception of picrotoxin (DMSO, 0.01% final bath concentration) all drugs were dissolved in purified water. Drugs were aliquoted at a concentration of 1/1000th of the final bath concentration and stored at −20 °C. Upon use, they were thawed and quickly dissolved in the recording medium.

### In vivo electrophysiology recordings

**Recording single-unit activity of prefrontal cortical neurons.** Extracellular single-unit activity was recorded using custom-built optrodes. An electrode bundle consisting of 32 individually isolated nichrome wires (12 μm diameter; 80% nickel, 20% chromium; Kanthal, Sandvik Material Technology, Germany) was glued to an optic fiber (Thorlabs, Germany, 200 μm diameter, 0.39NA, with the tip placed at 300–700 μm above the electrodes) to allow for optogenetic identification of mPFC-LHA cells. Electrodes were attached with gold pins to a 32-pin Omnetics connector using an electrode interface board (Open Ephys, www.openephys.org) which was mounted on a 3D printed holder (based on the Flexdrive[50], Open Ephys). The microwires were cut at an angle so that they were organized in a vertical space of 200–400 μm, whereafter they were gold-plated (Neuralynx, United States) to an impedance of 100 kΩ using a NanoZ impedance tester (Plexon, United States). The connector was grounded through a bare platinum-iridium wire (127 μm diameter; 90% platinum, 10% iridium; Science Products, Germany) placed on the cerebellum. Analog signals were acquired using a 32-channel RHD2132 headstage (Intan Technologies, USA), bandpass filtered between 2.5 and 7603.88 Hz, and digitized at 30 kHz. Data were transferred to the Open Ephys acquisition board using an Intan Technologies Serial Peripheral Interface cable (2.9 mm, interference of the cable during behavior was reduced by counterbalancing the weight of the cable with flexibly moving counterweights) and acquired by the Open Ephys GUI interface (version 0.5.2.2; https://open-ephys.org/gui)[51]. Behavior of the mice was recorded from above the cage using a Blackfly S USB camera (BFS-U3-31S4M-C, Teledyne FLIR), which was synchronized with the electrophysiological data by sending a TTL pulse for every acquired frame (sample rate 30 Hz) to the Open Ephys I/O board. The mice were also recorded from the front view using a webcam (HD C270, Logitech,

Switzerland). This data was synchronized by a light signal generated at a fixed interval by an Arduino-controlled LED, whose light signal was duplicated as a TTL output to the I/O board.

**Spike sorting.** Before spike-sorting, a common average reference https://github.com/cortex-lab/spikes/blob/master/preprocessing/applyCARtoDat.m[52] was applied to remove common noise contamination across channels. Spikes were then automatically detected and assigned to clusters using the template-matching algorithm of kilosort (version 2.0; https://github.com/MouseLand/Kilosort/releases/tag/v2.0) with the following settings: Ops.Th [10 4], Ops.AUCsplit [0.9], ops.Lam [10], ops.minFR [0.1]. Clustering results were manually checked for waveform shape, stable amplitude over time, and the presence of a clear refractory period. Moreover, resembling clusters were assessed to see if nearby clusters belonged to the same cell and were merged when appropriate. Data checking and curation were done in Phy (https://github.com/cortex-lab/phy). After curations in Phy, the isolation distance was calculated as a final quality control. Single units with an isolation distance of >20 were included in the dataset[53,54] (Average isolation distance of optotagged mPFC-LHA cells = 38.9, Minimum = 20.3, Maximum = 115.2).

**Optogenetic identification of mPFC-LHA neurons.** After every behavioral session, we applied short light pulses (2 ms) from a 473 nm DPSS laser (CNI Laser, China) with a random 6-9 s interval for ~30 min (over 200 trials), triggered using a microcontroller (Arduino, Italy)) and ~5-8 mW output at the fiber tip above the recording site. Depending on the presence of a voltage-volley during laser stimulation, a window of 1 or 2 ms after the laser onset was blanked. Significant light-induced activation was determined with the stimulus-associated spike latency test (SALT, $p < 0.01$; http://kepecslab.cshl.edu/salt.m)[53]. In brief, SALT compares the distribution of first spike latencies relative to the light pulse to epochs of the same duration in the stimulus-free baseline period. Additional criteria were included to prevent inclusion of false-positive neurons through network-related activity: a peak z-score>1.65, average first spike latency <7 ms, jitter <3.5 ms, reliability > 0.03, Pearson's correlation between the average waveform of a cluster and the average waveform of the first, putatively light-evoked spikes > 0.8, and a waveform duration above 550 microseconds (through-to-peak). Similar criteria have been used for mPFC-LHA opto-tagging, and ex vivo data of mPFC-LHA neurons suggest that a first spike latency below 8 ms limits false-positive identification of non-expressing neurons through recurrent connectivity[20] and the same criteria were also used in other mPFC opto-tagging studies[53,55,56]. According to these criteria, mPFC-LHA neurons could be photo-identified in 12 out of 16 mice that had correct placement of the optrode and proper expression of channelrhodopsin. All broad-spiking neurons that were not positively identified were categorized as non-identified mPFC neurons. To enhance the quality of comparison between opto-tagged and non-identified neurons, we only used non-identified neurons for analysis if in the same mice, opto-tagged neurons were present.

### In vivo fiber photometry recordings

We used a 2-site photometry system (Doric lenses) to detect calcium and non-calcium-dependent dynamics (GFP and isobestic channel). Two light-emitting diode delivered blue (465 nm, 40 ± 1 μw in the brain) and purple (405 nm, 20 ± 1 μw in the brain) light, which was guided to the mouse via a mini cube (ilFMC5-G2_IE(400-410)_E(460-490)_F(500-540)_O(580-680)_S) and optical patch cord (ø400 μm). Emitted light was detected with the mini-cube integrated photodetectors and transmitted to the console with different reference frequencies (GFP 572 Hz, ISO 209 Hz). The photometry system was controlled by Doric Studio software (version 6.3.2.0), which recorded 243 samples per second (50 times decimated). The relative changes in light (ΔF/F) were calculated with custom-made Python scripts (version

3.13.5). Code accessible here: https://doi.org/10.5281/zenodo.18670225.

First, outliers were removed (>4 std aways from the mean signal of 9 minutes) and data was smoothed (per 100 data points). We calculated the ΔF/F for each trial with -10 to -2 seconds before the onset as a baseline. Next, the session average and peak signals were extracted.

## Behavioral paradigms
All behavioral manipulations and tests were performed during the light phase.

## Two-day Social stress paradigm
The social subordination stress protocol used was designed according to a resident intruder paradigm. The resident, a Swiss-CD1 male previously used for breeding, was housed in a large Makrolon cage (type IV, Tecniplast, Italy). An intruder, an experimental male mouse, was placed in this cage. Fighting time was tracked live and animals were allowed to fight for 20 s, twice a day for two days, one fighting session in the morning (8.30–11.00 a.m.) and one in the afternoon (16.00–18.30 p.m.). For the remainder of the day, animals were separated by a perforated transparent splitter, allowing sensory, but no physical interaction. The control animals were co-housed with a non-familiar C57B6J male mouse for two days, preventing physical interaction. This short stress paradigm has been shown not to change body weight in the first days after the stressful experience when food intake is measured[3]. Consistent with this, no difference in body weight was observed between the experimental groups in this study. The schematics on social stress (Fig. 2D and Supplementary Fig. 2C) were created in BioRender. Meye, F. (2026) https://BioRender.com/mula9s9.

## Limited access choice diet.
In the limited access choice diet paradigm, animals were exposed to fat (i.e., tallow; Blanc de Boeuf, Vandermoortele, 9.0 kcal/g) and regular chow for 2 h a day in a clean cage (Makrolon cage type IV, Tecniplast, Italy). Animals were habituated to the limited-access cage and fat, on at least 3 occasions, before the start of the experiment. Animals were subjected to stress as described above, and on the day after, fat intake was measured over a 2-h fat access period, starting between 9 a.m. and 11 a.m. Unlimited chow was provided for the remaining 22 h of the day. The Fig. 1 panel C schematic of the food choice paradigm was created in BioRender. Meye, F. (2026) https://BioRender.com/ayjpdtm.

## Real-time place preference/aversion test.
The conditioned place preference test consisted of two sessions separated over two days. The test cage (73 × 25 cm) consisted of two chambers (30 × 25 cm) clearly separated by a narrower middle area. The two chambers had different flooring (fine grid vs coarse grid) and different walls (striped vs black) to provide contextual discriminability. In the baseline session, mice were allowed to freely explore the test cage. During the test session (20 min total duration), one of the compartments (counterbalanced design) was paired with optogenetic stimulation using closed-loop optogenetics. After 10 min the stimulation site was paired with the other compartment, to limit the effects of a priori preference for one of the compartments. Mice were recorded by a webcam (HD C270, Logitech, Switzerland). Closed-loop optogenetics was established by automatically tracking mice in Bonsai (Open Ephys), which triggered the Master-8 pulse generator through an Arduino upon entering the stimulation chamber, in turn triggering the laser at the dedicated frequency as long as the mouse was present in the stimulation compartment. The position of the animal was tracked in the cage offline using Ethovision 9 (Noldus, Wageningen). The preference score was calculated by scoring the total time spent in the stimulation compartment divided by the total time spent in the stimulation and non-stimulated compartments. Cages were cleaned in between animals with 70% ethanol.

## Food intake video analysis.
Videos were taken from the side of the feeding cage with webcams (HD C270, Logitech, Switzerland) and acquired using the Windows Camera App or Bonsai. Feeding bout analysis was done manually and blindly for the virus type. Data were acquired over different batches, which were analyzed by an independent observer.

## Recordings during social stress.
For social stress recordings, mice were handled extensively to get used to fixation and getting used to being attached to the recording cable. During the day of social stress recordings, mice were put into an unfamiliar cage of a CD1 aggressor mouse, which was temporarily removed from its home cage. After 15 min the CD1 mouse was introduced into the cage, but physical contact was prohibited by a transparent barrier. After another 45 min the barrier was removed, and the mice were allowed to physically interact (fight) for a cumulative 20 s whereafter the barrier was put back. This period lasted on average more than 4 minutes. Hence, during the total stress exposure time, physical contact, which could introduce noise in the recording, was limited.

## Restraint stress.
This stressor was induced by restraining mice twice a day for 15 minutes each for two days. The mice were placed individually in a conical Falcon tube (50 mL) previously drilled with thirty holes. Control mice were handled and placed back in their home cage after a couple of minutes.

## Locomotor activity and time spent in food zones.
Mice's position and speed in the food cage were automatically tracked in real-time from camera images or offline from the acquired webcam videos using Ethovision (version 9.1; Noldus Technologies, The Netherlands). Food zones were 9×11 cm in size. Time in the food zone was determined by calculating the cumulative time spent in the food zone per block of 10 minutes. The effect of 5 Hz stimulation on time in the food zone was determined by calculating the difference in time spent in the food zone during the individual three ON-Blocks with the preceding OFF-block and consequently taking the average of these three values obtained for the separate ON-Blocks.

## In vivo circuit approaches during behavioral experiments
### In vivo optogenetic stimulation experiments
**Optogenetic stimulation of mPFC-LHA in stress-naïve mice.** Before every experiment, mice were extensively handled and habituated to the food-context, fat, and optogenetic cables for at least 2 weeks before the start of the experiment. Mice had an incubation time of at least 12 weeks after viral injection for this experiment. A single experiment consisted of three baseline days, an optogenetic stimulation day, followed by another non-stimulation day. Mice were also attached to patch cables on days when no stimulation was applied. Raw food intake was measured on all days after one hour in which the mice were able to freely feed on regular chow (3.61 kcal/g, CRM (E), 801730, Special Diets Services) or Tallow (Blanc de Boeuf, Vandemoortele, 9,0 kcal/g). On stimulation days, mice were recorded with a webcam from the top, to allow for later offline tracking of the position of the mice in the food cages using Ethovision (Noldus, Netherlands). In a subset of the experiments, feeding behavior was recorded from the side using webcams (HD C270, Logitech, Switzerland). Time of testing across days was kept as constant as possible. Different frequencies were tested on the same mice and in the order 5 Hz, 10 Hz, and 1 Hz, with one week without stimulation in between each stimulation experiment.

On stimulation days, mice were connected to optogenetic patch cables (Thorlabs, 200 µm, 0.39 NA, FT200UMT), which ended in a ceramic ferrule (230 µm bore size, Thorlabs or Precision Fiber Products) and connected to the implant through a sleeve (Precision Fiber Products). Cables were connected to the laser (473 nm DPSS laser, CNI

Laser, China) using an FC-connector (Thorlabs) and via a rotary joint (Doric lenses, Canada), which divided light intensity equally over the two hemispheres. Light intensity was set to 5 mW at the tip of the fiber above the LHA. Light pulses (5 ms duration) at indicated frequencies were applied during blocks of ten minutes using a Master-8 pulse stimulator (A.M.P.I., Jerusalem, Israel) and were alternated with non-stimulation blocks of similar length. Mice were single-housed, and food and water were present ad libitum in the home cage.

**Optogenetic stimulation of mPFC-LHA in control versus stressed mice.** Viral incubation time and experimental setup for optogenetic stimulation are similar to those described above. 5 Hz stimulation was applied in alternating OFF and ON blocks with a duration of ten minutes. Mock-stimulated animals were attached to patch cables and laser-stimulated, but light transmission was completely blocked in the connecting sleeve on the head of the animal. For this experiment in particular, mice were habituated to fat (three times) in the food context and to optogenetic cables in the test cage for two weeks. Baseline food intake was measured 4–5 days before exposure to stress. During this session, animals were similarly attached to patch cables, but no stimulation was applied. The social stress paradigm was performed as described[3]. C57BL6 mice were introduced into a cage (Makrolon cage, type IV, Tecniplast, Italy) of an aggressor mouse (CD1, Janvier) and subjected to 4 x 20 s of physical fight over the course of two days (one fight in the morning and one in the afternoon) after which they were separated by a transparent wall for the remainder of the time. The cage of the Swiss-CD1 male mice contained enrichment that was previously used in the home cage, while the cage part of the experimental C57BL6 mice was enriched with 3 tissues. Food and water were present ad libitum in the stress and home cage. Control mice were subjected to 2 days of cohousing with a conspecific in the same cage, but separated by a transparent wall. On the day after stress, mice were put directly from the stress-context into the food context and returned to their home cage afterwards. One mouse was excluded from analysis on the basis of being a large statistical outlier. Three mice were excluded due to technical problems (e.g., malfunctioning video recording). All mice were individually housed during the experiment and had ad libitum access to food and water.

**Optogenetic stimulation of mPFC$_{SOCIAL\ STRESS}$ ensemble projections to LHA.** Mice were extensively handled and received mock intraperitoneal injections a couple of days before being exposed to an acute social stress (20scs) and received tamoxifen i.p injection (25 mg/kg) three hours later. Control mice were placed in a new cage with another C57Bl6 mouse separated by a perforated wall and received tamoxifen 3 h later as well. After a minimum of 6 weeks, mice were handled and habituated to the food-context, fat, and optogenetic cables. After at least two baseline days in the 2-choice paradigm, mice were tested in this paradigm for one hour, during which there was 5 Hz stimulation, presented in a pattern of 3 s on (5 Hz stimulation), 1 s off.

**Optogenetic stimulation of the LHA$_{VGLUT2}$-periPVN pathway experiment in stress-naïve mice.** The experimental setup is similar to the mPFC$_{SOCIAL\ STRESS}$ ensemble-LHA pathway experiment, but the mice were only stimulated at 10 Hz (alternation of 3 s ON, 1 s OFF).

**In vivo chemogenetic inhibition experiments**
**Chemogenetic inhibition of the mPFC-LHA pathway in control versus stressed mice during food intake..** Mice were extensively handled and habituated to the food context (3 times) and fixation and mock intraperitoneal injections for at least two weeks. First, mice were tested for their baseline fat and chow intake in the 2-choice context (2 h). For this baseline test, they received an intraperitoneal saline injection (150 µl) to rule out a potential effect of stress from this procedure. Baseline measurements were done at least 4 days

before being subjected to stress to prevent any influence of palatable food intake on their susceptibility to stress. The 2day social stress paradigm was done as described above. CNO was made fresh on the day of testing and applied at a dose of 5 mg/kg, 45 minutes before being placed in the food context. Mice removed from the stress context were injected with CNO and placed back in their home cage with ad libitum food before being put in the food context. Similarly, for the acute stress experiment, mice received the CNO injection 45 minutes before being placed in the food context and 25 minutes prior to stress. All mice were individually housed during the experiment and had ad libitum access to food and water. Two mice were excluded due to technical problems.

**Chemogenetic inhibition of LHA$_{VGLUT2}$ neurons during optogenetic stimulation of mPFC-LHA.** Mice were extensively handled and habituated to the food-context, fat, and optogenetic cables and mock intraperitoneal injections for at least 2 weeks before the start of the experiment. The experimental setup is similar to that described above, as well as the dose of CNO (5 mg/kg) 45 minutes before being placed in the food context, except that the mice were not exposed to stress and will be optogenetically stimulated at 5 Hz (alternation ON/OFF every 10 minutes). To ensure that the food intake measured during this experiment is not due to the ceiling effect, on the last day, mice were deprived of food for 21 h and re-expose two hours to the 2-choice context.

***Chemogenetic inhibition of LHA$_{VGLUT2}$ neurons during stress.*** Mice were habituated to the food context several days prior to the experiment. Mice then either experienced (40 s; -5 min), on each of two consecutive days, social stress via a CD1 aggressor, or instead a control experience by co-housing with a novel conspecific. On both days, 45 minutes prior to the experience, the mice received an i.p. injection of 5 mg/kg CNO. Food intake was monitored the subsequent day for 2 hours.

**In vivo fiber photometric recordings during stress and feeding.** Mice were exposed to lard and chow in an operant chamber (ENV-008CT) for 90-minute sessions. Food contacts were detected via infrared beam breaks (ENV-254-FB) and recorded using MED Associates software (MED-IV). From these recordings, we extracted the total eating duration and the number of eating bouts. An eating bout was defined as a continuous period of food contact lasting at least one second, with successive contacts separated by less than one second considered part of the same bout. Once the mice were habituated to the eating procedure, we recorded their neural activity photometrically during eating under baseline conditions. For the post-stress eating recordings, we employed a crossover design in which the order of conditions was pseudo-randomly determined using RandoMice[57], accounting for signal strength, fiber efficiency, and initial recording time. During this protocol, mice were subjected to the two-day social defeat paradigm as described previously. On the day following the defeat sessions, photometric recordings were performed again while the mice were eating. A recovery period of two weeks was provided between stress and control sessions. To record mPFC–LHA activity during social stress, we first conducted a three-minute baseline recording in the subject's home cage to minimize stress and allow for photobleaching. The subject was then exposed to a CD1 aggressor mouse in the aggressor's home cage while simultaneous photometric and video recordings were obtained. The mice were allowed to engage in fighting, and recordings were terminated after 40 seconds of active fighting. Video recordings were used to manually score for aggressive behaviors, including slamming, biting, chasing, and aggressive standing, but not approach behaviors, with one-second temporal

precision. The offset of each fight was defined as the end of the interaction or, when applicable, the landing of the C57BL/6 mouse. Aggressive behaviors occurring less than one second apart were considered part of a single fighting bout.

## Histology and microscopy

For each recording, the virus injection sites, as well as the placement of optic fibers and/or the optrodes, were examined. Only data obtained from mice with correct placements were included. After experiments, mice were anesthetized with pentobarbital (75–100 mg/kg) and transcardially perfused with Phosphate-Buffered Saline (PBS) containing 4% paraformaldehyde (PFA). Brains were post-fixed in 4% PFA overnight and transferred to PBS. Before slicing, brains were embedded in 4% agarose to maintain structural integrity. Coronal sections of the entire brain were sliced at 60 μm using a vibratome (VT1000s, Leica, Rijswijk, The Netherlands). For patch clamp experiments, out of 272 mice, 27 were excluded on the basis of misinjection of viruses. For in vivo experiments, a total of 13 mice were excluded based on histological assessment. For histology, we used the Paxinos & Franklin mouse brain atlas (The Mouse Brain, in stereotaxic coordinates, 2nd ed, 2001, Academic Press).

## Immunohistochemical staining for histology experiments

To stain for Channelrhodopsin, hM4DGi or control mCherry virus, tissue sections were washed in PBS (0.01 M, pH = 7.4) three times for 10 min before incubating them for 2 h with blocking solution (3% BSA, 0.5% Triton-X in 1x PBS). The primary antibody, Rabbit Anti-RFP (Rockland, 1:1000 in blocking solution, Lot:46510) that targets mCherry, was administered to the tissue sections using ~400 ul per well. Incubation was carried out overnight at 4 degrees on a shaker to facilitate optimal antibody priming. The next day, tissue sections were washed three times with PBS before applying a secondary antibody (goat-anti-Rabbit Alexa-488, Abcam, 1:500, Lot:GR3375958-2, Lot:GR3442006-1). The sections were incubated with the secondary antibody for two hours before being washed in PBS again for 3 times. Finally, tissue slices were mounted on glass slides and embedded in FluorSave (Millipore, 345789, Amsterdam, Netherlands) and covered with cover glasses. The microscopy slides were allowed to air dry overnight at room temperature before being stored in a 4-degree environment to maintain sample integrity. For staining of CoChR-GFP, anti-GFP antibody was used (Aves, GFP-1020, 1:1000 in blocking solution) and goat-anti-chicken Alexa-568 (Abcam, 1:500) as a secondary antibody.

## Validation of the AAV approach to label LHA$_{MCH}$ and LHA$_{OREX}$ cells for patch clamp.

Fluorescent cells were co-filled during electrophysiology recording with internal solution and biocytin (2 mg/ml). After recording, slices stayed overnight with 4% PFA at 4 °C and were then transferred to PBS 0.01% NaAz. For immunolabeling, slices were first washed with NaAz with PBS solution three times for 10 minutes in phosphate-buffered saline (PBS), then blocked for 1 hour at room temperature (RT) with blocking solution (10% normal goat serum (NGS) and 5% Triton X-100 in PBS). Followed by overnight at 4 °C incubation in primary antibody solution (with 1° antibodies, 2% NGS and 0.4% Triton X-100 in PBS). The next day, slices were washed three time during 15 min in PBS, and incubated for three hours at RT with the 2° antibody solution (with 2° antibodies, 2% NGS, and 0.4% Triton X-100 in PBS). After, slices were washed three times for 10 minutes in PBS. For the Orexin cells, first antibody used was Rabbit anti-Orexin A (1:1000, Abcam), followed by the secondary antibodies Goat anti-Rabbit 405 (1:500, Abcam) and streptavidin-AF488 (1:500). For MCH cells, slices were stained with Rabbit anti-MCH (1:1000, Sigma-Aldrich, St. Louis, MO, USA) followed by Goat anti-Rabbit 488 (1:500, Abcam) and streptavidin-AF555 (1:500). Cells were imaged with a confocal microscope.

**Microscopy.** Slices were imaged using an EPI fluorescence microscope (Axio imager M2, Zeiss) or a confocal microscope (LSM880, Zeiss).

## Data analysis and statistics

Experimental data were analyzed with Ethovision (V9.0, Noldus, NL), GraphPad (v9.0, Dotmatics, UK), SPSS (v26, USA), PatchMaster (v2x78.2, HEKA, USA), Python (v3.13.5), Igor Pro-8 (Wavemetrics, USA), FIJI (v2.14), and Matlab (vR2023B/vR2024B, Mathworks, USA). Experimental sample sizes were predetermined using power calculations (requiring power >0.8). These power calculations were performed together between the researchers and the local animal welfare body (AWB Utrecht). These calculations were guided by published studies, in-house expertise, and experimental pilots. Mice were randomly assigned to experimental groups. Mice or data points were not excluded from analyses unless noted. Behavioral and physiological experiments were always performed in multiple batches of animals (≥2), ensuring also internal replicability. Bar graphs are always reported and represented as mean ± SEM, with single data points plotted (single cell for electrophysiology and single animal for behavioral experiments). When statistically appropriate, One-way ANOVAs, Two-way ANOVAs, Multi-way Repeated-measures ANOVAs, or t-tests were applied. Post hoc analyses were performed in case of significant omnibus ANOVA outcomes. In the event the data were not normally distributed, statistical testing of group differences was performed using the Mann-Whitney U or Kolmogorov-Smirnov test. Testing was always performed two-sided with α = 0.05. Complete outcome of statistical analysis is shown in Supplementary Data 1.

## Reporting summary

Further information on research design is available in the Nature Portfolio Reporting Summary linked to this article.

# Data availability

All detailed outcomes of the statistical test are provided in Supplementary Data 1. Source Data are provided with this paper. Other data files are available from the corresponding author on request. Source data are provided with this paper.

# Code availability

The location for the codes used for the analyses in the current study have been referenced in the Methods section of the manuscript.

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

## Acknowledgements

We thank Fabien Ducrocq, Salvatore Lecca, Manuel Mameli, and the entire Meye Lab for discussions and critical reading of the manuscript. We thank Zoë Bor and Luc Sangers for assistance in analysis, Jaimie Hak and Simone Duis for assistance with surgeries, Maarten Werkman and Doortje Knobbe for assistance with systemic injections, and Nicky van Kronenburg for assistance in mouse breeding. This work was supported by (to FJM) the ERC under the European Union's Horizon 2020 research and innovation programme (grant agreement 804089; ReCoDE), the NWO VIDI grant 203.102, and partially by the NWO Gravitation project BRAINSCAPES: A Roadmap from Neurogenetics to Neurobiology (024.004.012), an Amsterdam UMC Fellowship to DR, and a Marie-Curie Individual Fellowship (CoMPOSE, grant agreement 898036) to RBP.

## Author contributions

L.F.S. performed and analysed all ex vivo electrophysiological recordings. K.L.K. performed the fiber photometric recordings, together with L.F.S. L.F.S. performed and analysed the following behavioral experiments: opto-stimulation of $LHA_{VGLUT2}$-Peri-PVN, opto-stimulation of mPFC-$LHA_{VGLUT2}$, combining with chemogenetic inhibition, opto-stimulation of mPFC TRAP ensemble, and restraint stress exposure. For these experiments, L.F.S. was further supported by F.J.M., W.D., and D.R. R.B.P. performed and analysed all other behavioral experiments, supported by A.A.C.B., A.S.J.N., and R.H. R.B.P. performed and analysed in vivo electrophysiological recordings. L.F.S., R.B.P., IW.-D., and M.C.M.L. performed stereotactic surgeries. FJM designed the study with R.B.P. and L.F.S. F.J.M. wrote the manuscript with L.F.S., R.A.H.A., R.B.P., and the help of all other authors.

## Competing interests

The authors declare no competing interests.
