## [Transparent Peer Review file · Nature Communications]

A prefrontal cortex-lateral hypothalamus circuit controls stress-driven increased food intake

Corresponding Author: Dr Frank Meye

Version 0:

Reviewer comments:

Reviewer #1

(Remarks to the Author)

Stress has bidirectional effects on feeding behavior. In this manuscript, Supiot et al provide evidence indicating that the mPFC-LHA circuit is involved in short-term stress-driven hypophagia and long-term hyperphagia. This finding is of novelty. However, more data is required to clarify how one pathway modulates feeding in two opposite ways.

Main points

1. The authors found that the prefrontal neurons projecting to LHA respond to stress behavior, but there is no evidence indicating that these neurons are also involved in feeding behavior. Fiber photometry recording is suggested here to confirm that the mPFC-LHA circuit is activated, maybe differently, during short-term stress-driven hypophagia and the long-term hyperphagia.
2. In Fig.2, the authors observed that the acute social stress (causing hypophagia)-activated mPFC neurons preferentially target LHAvglut2 subsets. However, in Fig 6D, blocking mPFC-LHA circuit has no effect on stress induced hypophagia. So, what is the significance of this targeting? More experiments should be done to set up the causal relationship between mPFC-LHAvglut2 and acute stress-related hypophagia.
3. In Fig.3, the authors have shown that mPFC innervated all four LHA subpopulations. The acute stress-trapped mPFC neurons mainly send inputs to LHAvglut2 neurons, but not LHAvgat neurons, and maybe that is why the authors focused on LHAvglut2 neurons in the following experiments. However, they couldn't exclude the effect of LHAvgat neurons, because there is no evidence indicating that the acute stress-trapped neurons are those work during feeding. The authors should check the role of LHAvgat neurons in the stress-driven hyperphagia or confirm that acute stress-trapped prefrontal neurons have effects on feeding behaviors.
4. In Fig. 4 and supplemental Fig. 5, the author observed a weakening of mPFC-LHAvglut2 projection but enhanced release probability of mPFC synapses onto LHAvglut2-VTA. These phenomena are interesting, but which one is the major cause for stress-induced hyperphagia? The authors should design more experiment to test it. For example, specifically block the mPFC-LHAvglut2-VTA projection and check its effect on stress-induced feeding behavior.

Minor comments

Fig 1A, What is the meaning of ">100" days?

Fig. 2. The number of neurons identified is too small. It is better to record neuronal activity via fiber photometry.

Fig.3F. The fluorescent intensity of the trapped neurons are too high. Are there any control experiments to confirm that the labeling is specific?

Fig3g. the n number is too low, are there only 2 neurons from the mice?

Fig 5C. the label C is missed. No description related to the statistical analysis between MCH group and control group.

Fig 6d. No description related to the statistical analysis between control group and stress group.

Reviewer #2

(Remarks to the Author)

The manuscript by Supiot et al. demonstrates the role of the mPFC-LHA pathway in stress-induced palatable food consumption. Using patch clamp recording and optogenetic/chemogenetic manipulations, the authors elucidate the synaptic mechanisms underlying stress-induced palatable food consumption. They found that mPFC projections to distinct LHA neurons modulate stress-induced fat overconsumption. Overall, this is an excellent study with a comprehensive analysis at multiple levels with novel findings. However, I believe that addressing the following concerns will improve the quality of this study.

1. It is well known that there is behavioral heterogeneity in mice exposed to social defeat stress (PMID: 17956738; 38228137): some mice show stress susceptibility while others do not (resilient). However, the authors did not mention this point at all.
2. The authors used only social defeat stress, which includes pain and injury. What about the effects of other types of stress? At least one other stressor including restraint stress or some other psychological stress model, should be required.
3. Most of the experiments are based on e-phys experiments and mechanistic insight is limited, so the causal relationship of the observed behaviors is still unclear. For example, the authors found that the activation of the mPFC-LHA(VGLUT2)-Peri-PVN pathway leads to increased PPR (Fig 4J), but its behavioral role is unclear. What happens when this pathway (and also the mPFC-LHA(VGLUT2)-VTA pathway) is stimulated/inhibited? Link to stress-induced hypophagia/hyperphagia? My feeling is that Figs 5 and 6 seem weak for a meaningful conclusion of this manuscript, because there is no in-depth LHA cell type information in these two experiments.
4. The authors selected four cell types in this study (VGLUT2, VGAT, OREX, and MCH). However, it would be appropriate for the authors to use an unbiased gene expression analysis (e.g., RoboTag-seq/scRNA-seq/Patch-seq) to better understand the LHA cell type associated with mPFC and/or PVN/VTA.
5. In Fig. S5, the authors used an anterograde transsynaptic AAV (AAV1-Cre). However, this was also able to spread retrogradely to presynaptic neurons (Zingg et al., Neuron 2017). Did the authors confirm and validate that there are no projections from the LHA to the mPFC?
6. I cannot find a validation experiment for the DREADD experiment. CNO treatment suppresses neuronal activity? Also, why did the authors use RFP-Alexa488 labeling instead of mCherry?

Reviewer #3

(Remarks to the Author)

Supiot and colleagues describe a neural pathway from prefrontal cortex to lateral hypothalamus that can control social stress-driven food intake. By combining optogenetic manipulations with electrophysiological recordings, the authors argue that the projection from the mPFC to LHA is critical for stress-driven fat intake and optogenetic stimulation of the mPFC-LHA increases fat intake. They also demonstrate that the mPFC-LHA network acutely responds to social stress and after social stress plasticity occurs specifically at mPFC synapses onto LHA glutamatergic (but not GABAergic) neurons. Finally, using optogenetics and chemogenetics, they argue that the mPFC-LHA pathway is sufficient and necessary for stress-driven increased fat intake.

While I am enthusiastic about the question and approach of the present manuscript, I struggle to link the disparate pieces of data into a cohesive explanation for the relationship between mPFC-LHA activity, stress-induced feeding, and synaptic plasticity. These various gaps should be filled before publication.

Major concerns:

1. The link between mPFC activity and LHAvglut or LHAvgat activity and behavior is not fully elucidated. The authors demonstrate that bulk activation of this pathway induces feeding and that mPFC neurons synapse onto both vglut2 and vgat neurons in the LHA. Since activation of LHA-vglut2 and LHA-vgat populations has opposite effects on food intake, how do the authors reconcile these results? One possibility would be to perform chemogenetic silencing of either LHAvglut2 or LHAvgat neurons during optogenetic stimulation of the mPFC-LHA pathway to determine which LHA population is contributing to the observed effects on feeding and/or stress-induced changes in feeding.

2. Relatedly, the observation that only 5-Hz, but not 1- or 10-Hz, stimulation increases fat intake is interesting, but should be linked via electrophysiology to LHA activity. For example, a prediction would be that LHAvglut2 and LHAvgat neurons exhibit divergent firing changes in response to mPFC stimulation (presumably, vgat neurons would be preferentially activated at 5Hz). The firing rates of LHAvglut2 and LHAvgat should be directly compared in response to trains of 1-, 5-, and 10-Hz stimulation to more convincingly link the behavior to LHA physiology. Presently, only data from 5Hz stimulation are shown in Fig. S3E. Based on those traces, it appears as if LHAvgat neurons fire with less fidelity than LHAvglut2 neurons. If this is true, and if increased firing of LHAvglut2 neurons is observed with higher frequencies of optogenetic stimulation, how can the authors reconcile the behavioral specificity of 5Hz stimulation with LHA activity?

3. The authors argue that the effects of mPFC stimulation are specific to fat intake, but it has not been demonstrated that the effect of PFC-LHA stimulation is specific to fat consumption. An alternative explanation is that pathway stimulation induces consumption of the most palatable or calorically dense food. Without additional experiments testing these possibilities, claims that stimulation is specific to fat (eg, Line 82) should be removed or qualified throughout. Would mPFC-LHA stimulation induce intake of chow if it were the only food available?

4. Are PFC-LHA neurons excited during food consumption? Bulk activation of this pathway induces food intake, but the neuronal responses indicate both excited and inhibited neurons during stress/post-stress epochs (Fig. 2). How do differently responding PFC-LHA neurons (increasing vs decreasing in response to acute stress) relate to downstream physiology or behavior? Linking the divergent responses of PFC neurons to behavior and LHA physiology would greatly strengthen the manuscript.

Minor:

5. Can PFC-LHA neurons follow 1-, 5-, and 10-Hz stimulation for extended time periods as used in the behavioral testing? The Barbano paper cited as justification that 5Hz stimulation induces specific behavioral outcomes was performed in a different cell type/brain region (LHAvgat-VTA projections) using a different opsin (Chr2) and shorter stimulation durations (5min). They also showed that both 5Hz and 10Hz stimulation increased food intake.

6. Data from non-phototagged neurons are not shown for Fig. 2. Do the PFC-LHA neurons represent a distinct population that is selectively modulated by acute social stress? Or is the distribution of response types of all recorded PFC neurons similar to that of the LHA-projecting subset? This is an important point to address with data or at least discuss as it would suggest either functional specificity or generality within the PFC neuron population.

7. Do individual mPFC neurons innervate both LHAvgat and LHAvglut2 neurons? Addressing this directly may be beyond the scope of the present manuscript, but determining whether the stress-excited and stress-inhibited neurons differentially target LHAvgat or LHAvglut2 populations would clarify much of the above-mentioned ambiguity. Some discussion of this possibility should be included.

Version 1:

Reviewer comments:

Reviewer #1

(Remarks to the Author)

Thank you for providing additional experimental data and clarifying the manuscript's focus on stress-driven hyperphagia. The new fiber photometry recordings and TRAP-based ensemble manipulations significantly strengthen the study. The authors have addressed my concerns

Reviewer #2

(Remarks to the Author)

The authors have addressed all of my concerns, and I have no additional comments.

Reviewer #3

(Remarks to the Author)

I commend the authors on a thorough revision which addresses all my previous concerns. I have no further comments.

Dear editor,

We thank the reviewers for their interest in the manuscript and for their valuable and constructive feedback. On the basis of their comments we have performed multiple new experiments, performed new analyses and rewrote certain parts of the manuscript to enhance focus. On the basis of this we have managed to further improve the manuscript. We specifically address the different points of the reviewers below.

REVIEWER #1

Stress has bidirectional effects on feeding behavior. In this manuscript, Supiot et al provide evidence indicating that the mPFC-LHA circuit is involved in short-term stress-driven hypophagia and long-term hyperphagia. This finding is of novelty. However, more data is required to clarify how one pathway modulates feeding in two opposite ways. We thank the reviewer for their evaluation of the novelty of the work. We understand their point related to hyperphagia versus hypophagia. As also further outlined below, we have now included several new datasets and analyses that clarify the importance of the mPFC-LHA pathway in mediating the *hyperphagic* aspects of stress. For a clearer focus of the manuscript as a whole we have therefore now chosen to also rewrite the manuscript more to clearly address the role of the pathway in specifically stress-driven hyperphagia.

Main points

1. The authors found that the prefrontal neurons projecting to LHA respond to stress behavior, but there is no evidence indicating that these neurons are also involved in feeding behavior. Fiber photometry recording is suggested here to confirm that the mPFC-LHA circuit is activated, maybe differently, during short-term stress-driven hypophagia and the long-term hyperphagia.

We appreciate this suggestion, and we have followed up on it. Specifically, we have now performed new fiber photometric recordings of the mPFC-LHA pathway (GCaMP). These data are shown in (New Main Figure 3E-J; New Suppl. Figure 3G-I). We found the following:

First, in accordance with our single-cell *in vivo* opto-tagged electrophysiology data, we now show with the fiber photometric recordings that the mPFC-LHA pathway shows a biphasic response to social stress: an initial increase and a subsequent dip in calcium activity.

We then performed calcium recordings when (non-stressed) animals were in the 2-choice food assay (with chow and fat at their disposal). We show that during

engagement with food, and in particular fat, there is a decreased activity in this pathway. After social stress, at a moment of increased fat intake drive, there is still a clear dip during fat intake in this mPFC-LHA network, although it is of similar size to that in non-stressed animals.

Together these new data indicate that the mPFC-LHA pathway is reactive both to stress and to fat consumption. They also suggest that the effects of stress in terms of plasticity in the pathway may occur more at the downstream synaptic level, where we indeed report them, rather than at the PFC somatic level (where the photometric measurements took place).

2. In Fig.2, the authors observed that the acute social stress (causing hypophagia)-activated mPFC neurons preferentially target LHAvglut2 subsets. However, in Fig 6D, blocking mPFC-LHA circuit has no effect on stress induced hypophagia. So, what is the significance of this targeting? More experiments should be done to set up the causal relationship between mPFC-LHAvglut2 and acute stress-related hypophagia.

Following the suggestion of the reviewer we have now added further experiments to investigate the role of the mPFC-LHA stress-sensitive ensemble in food intake. We now use TRAP-based ensemble tagging techniques to show that the mPFC neurons that are stress-responsive, and that project to mainly to glutamatergic LHA neurons, promote an increased fat intake when optogenetically stimulated (New Main Figure 7I-K), without affecting chow intake (New Suppl. Fig 7G). Together with other new findings on the importance of LHA glutamatergic neuron activity in producing stress-driven hyperphagia (see points below) this now more clearly positions the manuscript as an investigation of mPFC-LHA role in stress-hyperphagia.

3. In Fig.3, the authors have shown that mPFC innervated all four LHA subpopulations. The acute stress-trapped mPFC neurons mainly send inputs to LHAvglut2 neurons, but not LHAvgat neurons, and maybe that is why the authors focused on LHAvglut2 neurons in the following experiments. However, they couldn't exclude the effect of LHAvgat neurons, because there is no evidence indicating that the acute stress-trapped neurons are those work during feeding. The authors should check the role of LHAvgat neurons in the stress-driven hyperphagia or confirm that acute stress-trapped prefrontal neurons have effects on feeding behaviors.

In accordance with the suggestion of the reviewer we have confirmed that acute stress-trapped prefrontal neurons alter feeding behaviors, in particular driving food intake (New Main Figure 7I-K; see point 2 above). The ensemble-based experiments also show that during stress the mPFC directly stimulates glutamatergic LHA neurons.

4. In Fig. 4 and supplemental Fig. 5, the author observed a weakening of mPFC-LHAVglut2 projection but enhanced release probability of mPFC synapses onto LHAVglut2-VTA. These phenomena are interesting, but which one is the major cause for stress-induced hyperphagia? The authors should design more experiment to test it. For example, specifically block the mPFC-LHAVglut2-VTA projection and check its effect on stress-induced feeding behavior.

Indeed, in this study we observe that the mPFC-LHA network is multi-branched, with stress differentially affecting these branches (e.g. increasing, decreasing or not affecting synaptic communication within them). We added several new lines of evidence and discussion that further consolidate the ideas that stress both:

- a. Weakens excitatory mPFC synapses onto LHA_{VGLUT2}-peri-PVN neurons, thus reducing a brake on food intake (i.e. we added new connectivity analysis showing that most (~80%) LHA_{VGLUT2}—peri-PVN neurons receive excitatory mPFC input, and we also added new *in vivo* optogenetic data showing that stimulating the LHA_{VGLUT2}-peri-PVN pathway indeed reduces fat intake; (New Main Figure 6I-K)).
- b. Strengthens the mPFC synapses onto LHA_{VGLUT2}-VTA neurons, in part increasing a drive for food intake. Also here we provide new connectivity analysis showing that most (~80%) LHA_{VGLUT2}—VTA neurons receive excitatory mPFC input. This allows us to connect well with our previous study, which was fully focused on demonstrating that stress-driven potentiation of the LHA_{VGLUT2}→VTA pathway (which we now show is heavily innervated by mPFC) is crucial for the type of stress-driven hyperphagia we also investigated here with a similar behavioral paradigm (Linders et al., 2022; PMID: 36371405).

We now discuss both these new and previous findings more elaborately, proposing that it is likely through the recalibrated balance between multiple branches within the mPFC-LHA network, that stress drives the network as a whole more into a mode where its activity supports hyperphagic responses.

Minor comments

Fig 1A, What is the meaning of “>100” days?

For clarity we now describe the timelines in the methods section, and no longer in the figure panels themselves as this resulted in less clear situations in certain cases like here.

Fig. 2. The number of neurons identified is too small. It is better to record neuronal activity via fiber photometry.

We have now both increased the amount of included opto-tagged LHA-projecting mPFC

neurons, and we indeed (see point 1) complemented these data with population-level fiber photometric recordings.

Fig.3F. The fluorescent intensity of the trapped neurons are too high. Are there any control experiments to confirm that the labeling is specific?

We show that the expression is specific in several ways. Regarding the Cre-dependent virus (used for the TRAPing), we show that the virus itself is not being expressed in the mPFC if there is no Cre. We used for instance a Vglut2-Cre mouse which should not have Cre in the mPFC on account of those cell types mainly have Vglut1. Injecting the Cre-dependent virus in the mPFC indeed does not lead to expression of the virus (Suppl. Fig. 7E). Then in the TRAP2 mouse itself we show that a novel experience versus a social stress experience both leads to recruitment of mPFC cells but with very distinct connectivity profiles in LHA (Fig. 7H), further supporting specificity of experience-dependent targeting of distinct mPFC cells between social stress and novelty.

Fig3g. the n number is too low, are there only 2 neurons from the mice?

We now performed more experiments and increased the N and updated these numbers (Figure 7H).

Fig 5C. the label C is missed. No description related to the statistical analysis between MCH group and control group. Thank you. This is now corrected.

Fig 6d. No description related to the statistical analysis between control group and stress group. We thank the reviewer for pointing this out.

REVIEWER #2:

The manuscript by Supiot et al. demonstrates the role of the mPFC-LHA pathway in stress-induced palatable food consumption. Using patch clamp recording and optogenetic/chemogenetic manipulations, the authors elucidate the synaptic mechanisms underlying stress-induced palatable food consumption. They found that mPFC projections to distinct LHA neurons modulate stress-induced fat overconsumption. Overall, this is an excellent study with a comprehensive analysis at multiple levels with novel findings. However, I believe that addressing the following concerns will improve the quality of this study.

We thank the reviewer for their positive evaluation of our work.

1. It is well known that there is behavioral heterogeneity in mice exposed to social

defeat stress (PMID: 17956738; 38228137): some mice show stress susceptibility while others do not (resilient). However, the authors did not mention this point at all.

The reviewer raises an interesting point. Indeed, prior work has shown that chronic stress results at least in heterogeneity across mice regarding the extent of decreased social interaction and the amount of anhedonia that occurs. Whether such heterogeneity also occurs regarding the extent of increased intake of palatable food remains an intriguing possibility. Although there is variance in the response, we did not find an obvious behavioral dichotomy in stress-driven food intake in our study. Nor did we find such a clear dichotomy in our previous work (Linders et al., 2022; PMID: 36371405). We recognize that the separation may be more subtle than a dichotomy, as indicated in the papers indicated by the reviewer. Though our current study was not geared towards addressing the question of behavioral heterogeneity, we on the basis of the reviewer's suggestion have placed this in the Discussion as an important area of further research, also in the context of their suggested papers.

2. The authors used only social defeat stress, which includes pain and injury. What about the effects of other types of stress? At least one other stressor including restraint stress or some other psychological stress model, should be required.

We have followed up on the reviewer's suggestion. We have now performed new experiments with restraint stress, and we have now added these data to the updated manuscript. We show that restraint stress also increases the intake of palatable food (*New Supplementary Figure 6A-B*), and that this coincides with a weakening of mPFC input onto LHA_{VGLUT2} neurons, just as social stress does (*New Supplementary Figure 6C-E*). We also add further discussion on how this bolsters the idea that the effects we observe after social stress have more to do with the stress experience rather than other facets of that complex stimulus.

3. Most of the experiments are based on e-phys experiments and mechanistic insight is limited, so the causal relationship of the observed behaviors is still unclear. For example, the authors found that the activation of the mPFC-LHA(VGLUT2)-Peri-PVN pathway leads to increased PPR (Fig 4J), but its behavioral role is unclear. What happens when this pathway (and also the mPFC-LHA(VGLUT2)-VTA pathway) is stimulated/inhibited? Link to stress-induced hypophagia/hyperphagia? My feeling is that Figs 5 and 6 seem weak for a meaningful conclusion of this manuscript, because there is no in-depth LHA cell type information in these two experiments.

We have made several changes to the manuscript based on the reviewer's points:

(a) Regarding the question of linking mPFC-LHA_{VGLUT2}-Peri-PVN pathway function to its plasticity. We provide new behavioral validation that optogenetic stimulation of the mPFC-LHA(Vglut2)-Peri-PVN pathway indeed suppresses food intake (*New Suppl. Figure 6I-K*).

(b) We have altered the order of the figures such that aforementioned figure 6 (chemogenetic inhibition of the pathway as a whole to abolish stress-driven food intake) is no longer the final figure of the manuscript, but appears sooner (*New Figure 3A-D*).

Instead, we now added new data to indicate at the end of the work that the LHA glutamatergic neurons in particular, downstream of the mPFC neurons, are key integrators of stress and food reward intake (*New Figure 7*). Namely, we show that inhibition of LHA_{VGLUT2} neurons during social stress abolishes stress-driven hyperphagia (*New Figure 7A-C*). We then combined our previous ensemble-based connectivity experiments which showed that stress-activated mPFC neurons project more to glutamatergic than GABAergic LHA neurons (*Figure 7D-H*), with new *in vivo* optogenetic stimulation of this pathway. We show that this increases fat intake (*New Figure 7I-K*), directly linking the branch of the pathway that is stress-sensitive to palatable food intake promoting behaviors.

Together these changes provide further mechanistic links between physiology and behavior, and they result in an ending of the manuscript that goes more in-depth into the relevant LHA downstream targets of stress-modulated mPFC neurons for increasing food intake.

4. The authors selected four cell types in this study (VGLUT2, VGAT, OREX, and MCH). However, it would be appropriate for the authors to use an unbiased gene expression analysis (e.g., RoboTag-seq/scRNA-seq/Patch-seq) to better understand the LHA cell type associated with mPFC and/or PNV/VTA.

We agree that this would be a very interesting avenue to explore, but we hope that the reviewer agrees with us that while very interesting, this would in itself be a very extensive follow-up project that is not necessarily part of the scope of the current work. We do indeed intend to explore such avenues in the future.

5. In Fig. S5, the authors used an anterograde transsynaptic AAV (AAV1-Cre). However, this was also able to spread retrogradely to presynaptic neurons (Zingg et al., Neuron 2017). Did the authors confirm and validate that there are no projections from the LHA to the mPFC?

The reviewer is correct that this tracing strategy by itself should be taken with caution

regarding the direction of connectivity, indeed due to the ability of AAV to also partly retrogradely spread aside from its transsynaptic anterograde jump. We used the approach here solely as a qualitative screen, which we then followed up with actual functional circuit interrogation experiments (patch-clamp, retrotracing and optogenetics in slices) to confirm that these pathways flowing from mPFC downstream to LHA^{VGLUT2} neurons, and then further downstream to various other targets (VTA, Habenula, Peri-PVN), which came out of the tracing screen as potential targets, do indeed exist.

We now also better quantified the extent of connectivity between mPFC and these branches (Suppl. Fig. 6I). Finally, we changed the wording in the result section that dealt with the anatomical tracing experiment. For instance, earlier we had written that the tracing strategy “confirmed” the connection in itself, but this for abovementioned reasons is indeed not accurate and so we amended this now to say that it “suggests” it may be there (and which we then needed to test, and did so, with the optogenetic/patch strategy for confirmation).

6. I cannot find a validation experiment for the Dredd experiment. CNO treatment suppresses neuronal activity? Also, why did the authors use RFP-Alexa488 labeling instead of mCherry?

We now added new data to address this. We performed patch-clamp based validation of the chemogenetic inhibition strategy showing that indeed the mPFC-LHA pathway can be successfully chemogenetically inhibited with this approach (New Suppl. Fig. 3A).

We had used the RFP-Alexa488 labeling in that case as somewhat of an internal control to make sure that the staining itself had worked (as then the fluorescence linked to the staining is in another channel than the endogenous fluorescence that is also still there).

REVIEWER #3:

Supiot and colleagues describe a neural pathway from prefrontal cortex to lateral hypothalamus that can control social stress-driven food intake. By combining optogenetic manipulations with electrophysiological recordings, the authors argue that the projection from the mPFC to LHA is critical for stress-driven fat intake and optogenetic stimulation of the mPFC-LHA increases fat intake. They also demonstrate that the mPFC-LHA network acutely responds to social stress and after social stress plasticity occurs specifically at mPFC synapses onto LHA glutamatergic (but not GABAergic) neurons. Finally, using optogenetics and chemogenetics, they argue that the mPFC-LHA pathway is sufficient and necessary for stress-driven increased fat intake.

While I am enthusiastic about the question and approach of the present manuscript, I struggle to link the disparate pieces of data into a cohesive explanation for the

relationship between mPFC-LHA activity, stress-induced feeding, and synaptic plasticity. These various gaps should be filled before publication.

We thank the reviewer for their expressed enthusiasm for the work. We have performed further experiments and analyses to further connect the findings.

Main points:

1. The link between mPFC activity and LHAvglut or LHAvgat activity and behavior is not fully elucidated. The authors demonstrate that bulk activation of this pathway induces feeding and that mPFC neurons synapse onto both vglut2 and vgat neurons in the LHA. Since activation of LHA-vglut2 and LHA-vgat populations has opposite effects on food intake, how do the authors reconcile these results? One possibility would be to perform chemogenetic silencing of either LHAvglut2 or LHAvgat neurons during optogenetic stimulation of the mPFC-LHA pathway to determine which LHA population is contributing to the observed effects on feeding and/or stress-induced changes in feeding.

We thank the reviewer for their suggestion and we have performed several new experiments that demonstrate the important role played by LHA glutamatergic neurons in these effects.

1. We first performed a new experiment to show that chemogenetic inhibition of LHA glutamatergic neurons during social stress fully abolishes the subsequent stress-driven increase in fat intake that otherwise occurs (New Fig. 7A-C).

2. We then performed a new experiment in which we optogenetically manipulated the mPFC-LHA pathway either with LHA_{VGLUT2} neuronal activity left unaffected, or with their chemogenetic inhibition (New Suppl. Fig. 6A-D). We here replicated again that optogenetic stimulation by itself increased food intake. We reasoned that if the thus engaged LHA_{VGLUT2} branches would mainly serve to limit/counteract the food intake effect, that inhibiting them during pathway stimulation should unmask an even more potent intake of food. Instead, we observed that while mPFC-LHA pathway stimulation during LHA_{VGLUT2} inhibition increased food intake compared to no manipulation, it did not produce a stronger effect compared to just LHA_{VGLUT2} inhibition alone). We performed controls for a feeding ceiling effect, ensuring that it was possible for the animals to eat more in this timespan. This outcome suggests that activity of glutamatergic LHA neurons is not solely counteracting, but also indeed contributing also to the effect of mPFC-LHA network stimulation on food intake.

3. We had previously already performed an *ex vivo* connectivity experiment where we showed that stress-sensitive mPFC neurons (based on cFos-based TRAPing of stress-

ensembles) connected more to LHA glutamatergic than to GABAergic neurons. We have now also performed a new *in vivo* optogenetic experiment there as well (New Fig. 7I-K), showing that stimulating mPFC_{STRESS} ensembles with LHA glutamatergic downstream targets, was able to increase food intake.

Taken all together, these findings highlight the importance of activity of subsets of LHA glutamatergic neurons as a downstream integrator in the mPFC-LHA pathway that is crucial for (stress-driven) effects on food intake.

2. Relatedly, the observation that only 5-Hz, but not 1- or 10-Hz, stimulation increases fat intake is interesting, but should be linked via electrophysiology to LHA activity. For example, a prediction would be that LHAVglut2 and LHAVgat neurons exhibit divergent firing changes in response to mPFC stimulation (presumably, vgat neurons would be preferentially activated at 5Hz). The firing rates of LHAVglut2 and LHAVgat should be directly compared in response to trains of 1-, 5-, and 10-Hz stimulation to more convincingly link the behavior to LHA physiology. Presently, only data from 5Hz stimulation are shown in Fig. S3E. Based on those traces, it appears as if LHAVgat neurons fire with less fidelity than LHAVglut2 neurons. If this is true, and if increased firing of LHAVglut2 neurons is observed with higher frequencies of optogenetic stimulation, how can the authors reconcile the behavioral specificity of 5Hz stimulation with LHA activity?

We have now replaced the earlier singular example of how the cells responded to the optogenetic stimulation, with a more extensive investigation of the fidelity of synaptic responsivity over time at different stimulation rhythms. These data (New Suppl. Fig. 4J) show that for both LHAV_{GLUT2} and LHAV_{GAT} cells the synaptic fidelity remains high with 1 and 5 Hz stimulation, whereas it starts to deviate over time at 10 Hz. Continuous 10 Hz stimulation is reasonably well followed by LHAV_{GLUT2} neurons, but drops off more in LHAV_{GAT} neurons over time. It is therefore possible that especially more long-term 10 Hz stimulation as applied here, would produce a different behavioral effect than 5 Hz stimulation, as it maintains engagement of a different mixture of LHA cell types than does 5 Hz. This also means that 10 Hz stimulation may not be intrinsically unable to drive food intake, but would need to not be provided continuously to do so. We now discuss these aspects of the frequency-dependent findings and their caveats in more nuance than before in the Discussion.

3. The authors argue that the effects of mPFC stimulation are specific to fat intake, but it has not been demonstrated that the effect of PFC-LHA stimulation is specific to fat consumption. An alternative explanation is that pathway stimulation induces consumption of the most palatable or calorically dense food. Without additional experiments testing these possibilities, claims that stimulation is specific to fat

(eg, Line 82) should be removed or qualified throughout. Would mPFC-LHA stimulation induce intake of chow if it were the only food available?

The reviewer is correct. We have decided to alter the wordings of our claims in accordance with the reviewer's suggestion, and we add their potential alternative explanation as a point in the Discussion.

4. Are PFC-LHA neurons excited during food consumption? Bulk activation of this pathway induces food intake, but the neuronal responses indicate both excited and inhibited neurons during stress/post-stress epochs (Fig. 2). How do differently responding PFC-LHA neurons (increasing vs decreasing in response to acute stress) relate to downstream physiology or behavior? Linking the divergent responses of PFC neurons to behavior and LHA physiology would greatly strengthen the manuscript.

We have now added new experimental data using fiber photometry showing that mPFC-LHA neurons are at a population level transiently inhibited during food consumption (*New Main Figure 3E-J; New Suppl. Figure 3G-I*). At the same time we had shown that both bulk activation of the pathway induces food intake, and we also now provide new experimental data showing that the stress-sensitive ensemble of mPFC neurons and their projection the LHA also increase food intake (point 1; *New Fig. 7I-K*). These new efforts, including *in vivo* ensemble manipulation, offer a connection between stress-sensing neurons within the network and feeding responses.

We discuss multiple scenarios that can explain why mPFC-LHA activity dips during food intake, while pathway stimulation can increase food intake. In brief, one scenario draws inspiration from arcuate nucleus AgRP neurons, which are also potent drivers of food intake when stimulated, yet dip in their activity during food intake. For AgRP neurons, their activity may reflect an energy need to be fulfilled, which is then temporarily reduced during actual bouts of food intake. Similarly, it is possible that mPFC-LHA pathway activity reflects a conditional (e.g. under stress) need for the intake of high caloric (palatable) sources. Another scenario is that mPFC-LHA network activity produces a signal for continued engagement with the activity of food seeking. We discuss these potential interpretations.

Minor points:

5. Can PFC-LHA neurons follow 1-, 5-, and 10-Hz stimulation for extended time periods as used in the behavioral testing? The Barbano paper cited as justification that 5Hz stimulation induces specific behavioral outcomes was performed in a different cell type/brain region (LHAvgat-VTA projections) using a different opsin (ChR2) and shorter stimulation durations (5min). They also showed that both 5Hz and 10Hz stimulation

increased food intake.

We added new ex vivo patch clamp data to address this (see point 2; New Suppl. Fig. 4J). We show that during continuous optogenetic stimulation, both LHA_{VGAT} and LHA_{VGLUT2} neurons follow 1Hz and 5 Hz rhythms quite reliably over time. Instead, with continuous 10 Hz there is a drop off in fidelity over time in LHA_{VGAT} neurons, but not LHA_{VGLUT2} neurons. We have now added more nuance in our discussion of how distinct rhythms, both keeping in frequency and duration in mind, can have differential effects on mPFC-LHA network branch engagement and on resultant behaviors.

6. Data from non-phototagged neurons are not shown for Fig. 2. Do the PFC-LHA neurons represent a distinct population that is selectively modulated by acute social stress? Or is the distribution of response types of all recorded PFC neurons similar to that of the LHA-projecting subset? This is an important point to address with data or at least discuss as it would suggest either functional specificity or generality within the PFC neuron population.

We have now added these data as the reviewer suggested we do (New Suppl. Fig. 2I-J). We find that mPFC-LHA neurons are modestly enriched in terms of their proportion of modulation due to stress as compared to non-phototagged mPFC putative pyramidal neurons.

7. Do individual mPFC neurons innervate both LHA_{Vgat} and LHA_{Vglut2} neurons? Addressing this directly may be beyond the scope of the present manuscript, but determining whether the stress-excited and stress-inhibited neurons differentially target LHA_{Vgat} or LHA_{Vglut2} populations would clarify much of the above-mentioned ambiguity. Some discussion of this possibility should be included.

This is indeed an interesting point. We now discuss whether this is likely. Based on our ensemble-tagging approach, where we manage to express optogenetic actuators in subsets of mPFC neurons, we show that this can result in synaptic connectivity with certain LHA neuronal cell types, without also causing synaptic responses in others. For instance, the stress-responsive mPFC ensemble connects to glutamatergic LHA neurons but not GABAergic. This suggests that it is very likely that there is at least partial segregation of mPFC neurons projecting to glutamatergic or GABAergic LHA neurons. We point out in the discussion that elucidating these mPFC identities is an important follow-up that needs to be performed.